# Non-reciprocity across scales in active mixtures

Alberto Dinelli[1], Jérémy O'Byrne [1,2], Agnese Curatolo[3], Yongfeng Zhao [4], Peter Sollich [5,6] & Julien Tailleur [1,7] ✉

In active matter, particles typically experience mediated interactions, which are not constrained by Newton's third law and are therefore generically non-reciprocal. Non-reciprocity leads to a rich set of emerging behaviors that are hard to account for starting from the microscopic scale, due to the absence of a generic theoretical framework out of equilibrium. Here we consider bacterial mixtures that interact via mediated, non-reciprocal interactions (NRI) like quorum-sensing and chemotaxis. By explicitly relating microscopic and macroscopic dynamics, we show that, under conditions that we derive explicitly, non-reciprocity may fade upon coarse-graining, leading to large-scale equilibrium descriptions. In turn, this allows us to account quantitatively, and without fitting parameters, for the rich behaviors observed in microscopic simulations including phase separation, demixing, and multi-phase coexistence. We also derive the condition under which non-reciprocity survives coarse-graining, leading to a wealth of dynamical patterns. Again, our analytical approach allows us to predict the phase diagram of the system starting from its microscopic description. All in all, our work demonstrates that the fate of non-reciprocity across scales is a subtle and important question.

Our ability to design and engineer new materials largely relies on the possibility to infer their large-scale properties from their microscopic constituents. For equilibrium systems, statistical mechanics allows us to do so by relating the macroscopic free energy to the microscopic partition function and the Boltzmann weight. As a result, the emerging properties of equilibrium systems can be predicted by balancing energy and entropy. This general principle, at the root of so many industrial innovations over the past century, comes with a strong restriction: it only applies to the steady state of systems satisfying detailed balance, thus excluding the vast class of nonequilibrium systems and transient dynamical phenomena. An important challenge is thus to develop theoretical frameworks that would allow us to relate the microscopic description of nonequilibrium systems to their emerging behavior.

This is particularly important for active systems, which comprise large assemblies of individual units able to exert non-conservative forces on their environment[1]. From spontaneously flowing matter[2–5] to living crystals[6–10], active materials display phases without counterparts in equilibrium physics[11]. This rich phenomenology relies in part on the existence of non-reciprocal interactions (NRI) between active particles, which have attracted a lot of attention recently. From the spontaneous emergence of traveling waves to anomalous mechanics and odd elasticity, NRI have indeed been shown to lead to a wealth of exciting phenomena[12–25].

[1]Université Paris Cité, Laboratoire Matière et Systèmes Complexes (MSC), UMR 7057 CNRS, F-75205 Paris, France. [2]Department of Applied Maths and Theoretical Physics, University of Cambridge, Centre for Mathematical Sciences, Wilberforce Rd, Cambridge CB3 0WA, UK. [3]John A. Paulson School of Engineering and Applied Sciences and Kavli Institute for Bionano Science and Technology, Harvard University, Cambridge, MA 02138, USA. [4]Center for Soft Condensed Matter Physics and Interdisciplinary Research & School of Physical Science and Technology, Soochow University, 215006 Suzhou, China. [5]Institute for Theoretical Physics, Georg-August-Universität Göttingen, 37 077 Göttingen, Germany. [6]Department of Mathematics, King's College London, London WC2R 2LS, UK. [7]Department of Physics, Massachusetts Institute of Technology, Cambridge, MA 02139, USA. ✉e-mail: jgt@mit.edu

In the simplest case of systems with pairwise forces, NRI correspond to the breakdown of Newton's third law[12,24,26], which states that if particle $i$ exerts a force $\mathbf{f}_{ij}$ onto particle $j$ then $\mathbf{f}_{ji} = -\mathbf{f}_{ij}$. Such pairwise forces are an idealized limit for most active particles: experimental systems instead typically involve complex mediated $N$-body interactions like chemotaxis, quorum sensing, or hydrodynamic interactions that need not be reciprocal. In all cases, predicting how such microscopic interactions impact the emerging behavior is a challenging, indeed mostly impossible task (note that, for some passive systems, it has been shown that non-reciprocal interactions may allow for a conservation law of a generalized momentum-like quantity[26]). An appealing alternative has recently been proposed: to postulate phenomenological theories in which action-reaction is directly broken at the macroscopic scale[16,17]. The analysis of the large-scale behavior then amounts to a non-linear dynamics problem for which a wealth of tools are available[16,17,22,23,27–31]. However, a major limitation is that, in the presence of NRI, there is no generic way to infer which microscopic systems correspond to a given macroscopic description. This not only prevents us from assessing the scope of these theories, but it also deprives us of guiding principles when it comes to engineering microscopic active systems to realize the exciting emerging behaviors observed at the macroscopic scale.

In this article, we bridge the gap between microscopic and macroscopic descriptions of active systems with non-reciprocal interactions, which allows us to show that the violation of action-reaction is strongly scale-dependent. To do so, we study active mixtures, which comprise several types of interacting active particles and have attracted a lot of interest recently[12,16,17,21–24,32–44]. We first consider active particles that interact via quorum sensing (QS), i.e., regulate their motility according to the local density of their peers. QS is generic in nature[45], where it is typically mediated by diffusing signalling molecules. For microorganisms, it plays an important role in regulating diverse biological functions, from bioluminescence[46–49] and virulence[50] to biofilm formation[51] and swarming[52]. Furthermore, QS can also be engineered in the lab, for instance using light-controlled self-propelled colloids[53–55]. We then close this article by showing that our results generalize to chemotactic interactions and by discussing the case of pairwise forces.

## Results

We consider $N$ 'species' of active particles and denote by $\rho_\mu(\mathbf{r})$ the density field of species $\mu$. For concreteness, we present our results for active Brownian particles (ABPs)[56] and run-and-tumble particles (RTPs)[57], but we stress that they hold more generally and also apply, for instance, to active Ornstein-Uhlenbeck particles[58–60]. QS interactions between the species then lead to the following dynamics for particle $i$ of species $\mu$:

$$\dot{\mathbf{r}}_{i,\mu} = v_\mu(\mathbf{r}_{i,\mu}, [\{\rho_\nu\}])\mathbf{u}_{i,\mu}, \tag{1}$$

where the self-propulsion speed $v_\mu$ is both a function of $\mathbf{r}_{i,\mu}$ and a functional of all density fields. The particle orientation $\mathbf{u}_{i,\mu}$ is a unit vector that undergoes either rotational diffusion (ABPs) or tumbles instantaneously (RTPs), with a persistence time $\tau_\mu$ (note that making $\tau_{i,\mu}$ a function of $\mathbf{r}_i$ and a functional of $\{\rho_\nu\}$ does not lead to any interesting phenomenology so, for simplicity, we do not consider this case). We note that Eq. (1) is generically non-reciprocal. To see this, we split the self-propulsion speed $v_\mu$ between a bare contribution $\bar{v}_\mu \equiv v_\mu(\mathbf{r}_{i,\mu}, [\{\rho_\nu = 0\}])$ and an interaction term $\Delta v_\mu(\mathbf{r}_{i,\mu}, [\{\rho_\nu\}]) \equiv v_\mu(\mathbf{r}_{i,\mu}, [\{\rho_\nu\}]) - \bar{v}_\mu$, which allows us to rewrite Eq. (1) as:

$$\dot{\mathbf{r}}_{i,\mu} = \bar{v}_\mu \mathbf{u}_{i,\mu} + \mathbf{f}_{i,\mu}, \qquad \mathbf{f}_{i,\mu} = \Delta v_\mu(\mathbf{r}_{i,\mu}, [\{\rho_\nu\}])\mathbf{u}_{i,\mu}. \tag{2}$$

The effective $N$-body interactions $\mathbf{f}_{i,\mu}$ are clearly non-reciprocal as can easily be checked by considering a system with two particles: $\mathbf{f}_{i,\mu}$ and $\mathbf{f}_{j,\nu}$ are along $\mathbf{u}_{i,\mu}$ and $\mathbf{u}_{j,\nu}$, and thus generically not even collinear (see Supplementary Movie 1). To address how this non-reciprocity affects the large-scale properties, we first coarse-grain the dynamics (1) into a fluctuating hydrodynamic description that captures the large-scale long-time dynamics of the system. We then compute the steady-state entropy production rate of the resulting fluctuating hyrodynamics and show that it vanishes whenever

$$\frac{\delta \log v_\mu(\mathbf{r})}{\delta \rho_\nu(\mathbf{r}')} = \frac{\delta \log v_\nu(\mathbf{r}')}{\delta \rho_\mu(\mathbf{r})} \qquad \text{for any } \mu, \nu. \tag{3}$$

In this case, the microscopic non-equilibrium dynamics leads to an effective large-scale equilibrium dynamics and non-reciprocity vanishes upon coarse-graining. Condition (3) can thus be seen as the first non-trivial generalization of Newton's action-reaction principle to a microscopic model of active mixtures in the presence of many-body mediated interactions. We show that the system then admits an effective free energy which allows us to predict its emerging behavior. Remarkably, this allows us to construct the phase diagram of the system from its microscopic dynamics without any fit parameters, a rare achievement even in equilibrium. On the contrary, when Eq. (3) is violated, non-reciprocity survives coarse-graining and we derive a sufficient condition on the microscopic dynamics to observe travelling patterns that explicitly break time-reversal symmetry at the macroscopic scale. All in all, our work thus demonstrates that the fate of non-reciprocity across scales is a subtle and important question. To support the generality of this statement, we close our article by extending our results to chemotactic interactions where macroscopic reciprocity may again emerge despite microscopic NRI.

### Fluctuating hydrodynamics

We start by coarse-graining the microscopic dynamics (1) in the presence of QS interactions. The hydrodynamic modes are the fluctuating conserved density fields $\rho_\mu(\mathbf{r}) = \sum_j \delta(\mathbf{r} - \mathbf{r}_{j,\mu})$. As shown in the Supplementary Information, the stochastic dynamics of the density fields in $d$ space dimensions can be obtained as $N$ coupled Itō-Langevin equations:

$$\partial_t \rho_\mu = -\nabla_\mathbf{r} \cdot \left[ \mathbf{V}_\mu \rho_\mu - D_\mu \nabla_\mathbf{r} \rho_\mu + \sqrt{2D_\mu \rho_\mu}\, \boldsymbol{\Lambda}_\mu \right], \tag{4}$$

where the $\boldsymbol{\Lambda}_\mu(\mathbf{r}, t)$ are independent Gaussian white noise fields of zero mean, unit variance, and independent components. The collective diffusivities and drifts then read

$$D_\mu = \frac{v_\mu^2(\mathbf{r}, [\{\rho_\nu\}])\tau_\mu}{d}, \qquad \mathbf{V}_\mu = -D_\mu \nabla \log[v_\mu(\mathbf{r}, [\{\rho_\nu\}])]. \tag{5}$$

The fluctuating hydrodynamics (4) describes the large-scale dynamics of the density fields over time and space scales much larger than the microscopic persistence time and length. It can conveniently be rewritten as

$$\partial_t \rho_\mu = \nabla_\mathbf{r} \cdot \left[ M_\mu \nabla \mathrm{u}_\mu + \sqrt{2M_\mu}\, \boldsymbol{\Lambda}_\mu \right], \tag{6}$$

where $M_\mu(\mathbf{r}) = \rho_\mu(\mathbf{r}) D_\mu(\mathbf{r}, [\{\rho_\nu\}])$ is a density-dependent collective mobility and $\mathrm{u}_\mu(\mathbf{r}) = \log[\rho_\mu(\mathbf{r})] + \log[v_\mu(\mathbf{r}, [\{\rho_\nu\}])]$ is an effective chemical potential. One can then study the large-scale behavior of the system using Eq. (6) and connect it to the microscopic dynamics using Eq. (5).

### When non-reciprocity vanishes upon coarse-graining

Note that the fluctuating hydrodynamics (6) takes a form reminiscent of $N$ coupled model-B dynamics[61] and it is thus natural to ask whether we have coarse-grained our microscopic dynamics into an effective

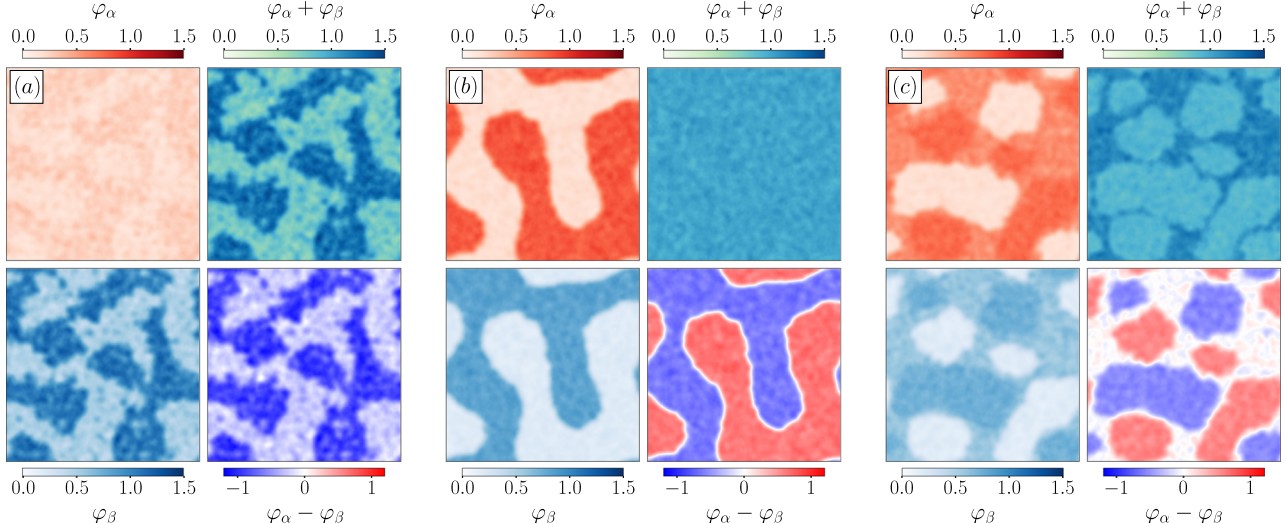

**Fig. 1 | Simulations of two species of RTPs with self-inhibition and global activation of the motility.** The self-propulsion speed is regulated through Eq. (11). The figures report simulation results for different average densities $\rho_{\alpha,\beta}^0$. Normalized densities are defined as: $\varphi_\mu = \rho_\mu/(\rho_\alpha^0 + \rho_\beta^0)$. **a** For $\rho_\alpha^0 = 15$ and $\rho_\beta^0 = 50$, species $\beta$ undergoes phase separation (bottom left), accompanied by a slight modulation of the density of $\alpha$ (top left). The total density field is inhomogeneous (top right). The local difference in composition simply recapitulates the variations of species $\beta$ (bottom right). **b** For $\rho_\alpha^0 = \rho_\beta^0 = 55$, the two species demix: the total density field is homogeneous (top right) while each species experiences phase separation (left column). The dense phase of one species is co-localized with the dilute phase of the other species (bottom right). **c** For $\rho_\alpha^0 = \rho_\beta^0 = 75$, we observe a triple-coexistence regime. Each species exhibits dense, dilute, and intermediate phases (left column). The intermediate phases of both species are co-localized, whereas their dense and dilute phases are demixed (bottom right). The overall density field shows the existence of a well-mixed dense phase---which results from the sum of the intermediate densities of each species---and one with an intermediate density---which corresponds to the demixed regions (top right). All parameters and numerical details are given in Methods.

equilibrium one. To address this question, we show in the Supplementary Information that the stochastic dynamics (6) leads to a steady-state entropy production rate given by[62–64]:

$$\sigma \equiv \lim_{t_f \to \infty} \frac{1}{t_f} \log \frac{\mathcal{P}[\{\rho_\nu(\mathbf{r},t)\}; t_f]}{\mathcal{P}^\dagger[\{\rho_\nu(\mathbf{r},t)\}; t_f]} = \int d^d\mathbf{r} \sum_{\mu=1}^{N} \left\langle M_\mu \left[ \nabla \left( \mathfrak{u}_\mu + \frac{\delta \log P_s}{\delta \rho_\mu(\mathbf{r})} \right) \right]^2 \right\rangle, \quad (7)$$

where $\mathcal{P}[\{\rho_\nu\}; t_f]$ is the probability of observing a trajectory $\{\rho_\nu(\mathbf{r},t)\}$ in a time window $(0, t_f)$, $\mathcal{P}^\dagger$ is the probability of the time-reversed trajectory, and $P_s[\{\rho_\nu\}]$ is the steady-state distribution. We stress that $\sigma$ characterizes the production of entropy in the steady state due to the irreversibility of the coarse-grained dynamics, and not that due to the relaxation of an initial measure $P_0[\{\rho_\nu\}]$ into the steady state $P_s[\{\rho_\nu\}]$[62–64]. When $\sigma$ is positive, the large-scale dynamics of the system in not symmetric under time-reversal and it converges toward a macroscopically-out-of-equilibrium steady steate (note that, at this coarse-grained scale, all connections to heat and thermodynamics are lost and $\sigma$ is simply a measure of irreversibility[1,64–67]). Equation (7) thus relates the irreversibility of the system at the macroscopic scale to its microscopic parameters $\{v_\mu\}$ through $\mathfrak{u}_\mu = \log v_\mu + \log \rho_\mu$. For generic QS interactions, $\sigma$ is positive and the coarse-grained dynamics is out of equilibrium. However, the model admits a macroscopic equilibrium limit whenever there exists a functional $\mathcal{F}[\{\rho_\nu\}]$ such that $\mathfrak{u}_\mu(\mathbf{r}) = \frac{\delta \mathcal{F}}{\delta \rho_\mu(\mathbf{r})}$. It is then easy to check that $P_s[\{\rho_\nu\}] \propto \exp[-\mathcal{F}[\{\rho_\nu\}]]$ and that $\sigma$ vanishes: the fluctuating hydrodynamics of the system is then a bona-fide equilibrium dynamics that satisfies detailed balance with respect to $P_s$. Furthermore, $\mathcal{F}$ is mathematically equivalent to a Landau free energy in equilibrium and plays the role of a Lyapunov functional for the dynamics, since $\partial_t\langle \mathcal{F} \rangle = -\int d\mathbf{r} \sum_\mu \langle M_\mu (\nabla \frac{\delta \mathcal{F}}{\delta \rho_\mu})^2 \rangle < 0$. This shows that the dynamics relaxes toward density profiles that minimize the Landau free energy $\mathcal{F}$, hence allowing us to determine the most probable configurations of the system using a variational principle.

To assess whether such an effective free energy exists, we need to determine the conditions under which $\mathfrak{u}_\mu$ can be written as a functional derivative. To do so, we generalize the functional Schwarz theorem[1] to the case of $N$ coupled stochastic field equations. As shown in the

Supplementary Information, this leads to a system of $N^2$ equations in the sense of distributions:

$$\forall(\mu,\nu), \qquad \mathcal{D}_{\mu\nu}(\mathbf{r},\mathbf{r}') \equiv \frac{\delta \mathfrak{u}_\mu(\mathbf{r})}{\delta \rho_\nu(\mathbf{r}')} - \frac{\delta \mathfrak{u}_\nu(\mathbf{r}')}{\delta \rho_\mu(\mathbf{r})} = 0. \quad (8)$$

Using the explicit expression for the chemical potential $\mathfrak{u}_\mu$ then directly leads to the condition (3) for the microscopic dynamics. When the self-propulsion depends solely on local densities, i.e., $v_\mu(\mathbf{r}) \equiv v_\mu(\rho_1(\mathbf{r}), ..., \rho_N(\mathbf{r}))$, Eq. (8) simplifies into

$$\frac{\partial \log v_\mu}{\partial \rho_\nu} = \frac{\partial \log v_\nu}{\partial \rho_\mu}, \quad (9)$$

whose full solution space can be constructed explicitly. Indeed, the solutions to Eq. (9) are generated by the gradient in $\rho_\mu$-space of all the 'potentials' $U(\rho_1, ..., \rho_N)$ through $\log v_\mu = \frac{\partial U}{\partial \rho_\mu}$. The effective free energy can then be directly computed as

$$\mathcal{F}[\{\rho_\mu\}] = \int d\mathbf{r} f(\mathbf{r}) \quad (10a)$$

$$f(\mathbf{r}) = U(\{\rho_\mu(\mathbf{r})\}) + \sum_\mu \rho_\mu(\mathbf{r}) \log \rho_\mu(\mathbf{r}). \quad (10b)$$

Importantly, even though the microscopic dynamics then maps onto a bona-fide equilibrium problem at the macroscopic scale, Eq. (2) still holds and the contribution of particle $(i,\mu)$ to the velocity $\dot{\mathbf{r}}_{j,\nu}$ of particle $(j,\nu)$ is not equal and opposite to the contribution of particle $(j,\nu)$ to $\dot{\mathbf{r}}_{i,\mu}$. In other words, momentum conservation is still violated at the microscopic scale and the microscopic interactions are still non-reciprocal. Equation (9) can then be seen as a non-trivial microscopic constraint on the QS interactions such that action-reaction is restored at the macroscopic scale. In this sense, we say that it generalizes Newton's third law to quorum-sensing interactions.

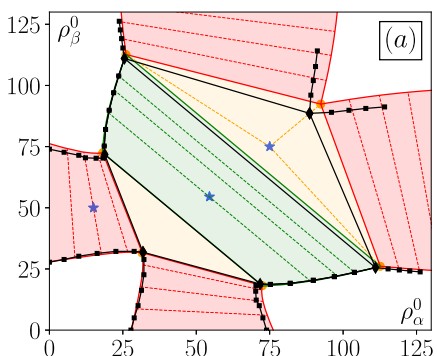
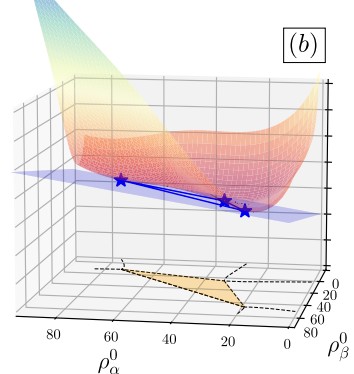

**Fig. 2 | Phase diagram and common-tangent construction in the presence of self-inhibition and global activation of motility. a** Phase diagram of two species of RTPs experiencing self-inhibition and global activation of motility according to Eq. (11). White regions correspond to homogeneous well-mixed phases. Red, green, and ochre regions indicate one-species phase separation, demixing, and triple phase coexistence, respectively. Stars correspond to snapshots shown in Fig. 1a–c.

Coexistence lines (solid) and tie-lines (dashed) are predicted using a tangent plane construction on the free energy density $f(\rho_\alpha, \rho_\beta)$ as detailed in the Supplementary Information. Black squares show coexisting densities measured in simulations. **b** Plot of the free energy density in the triple coexistence regime from Fig. 1c. The points where the tangent plane in blue touches the surface determine the three compositions that will be observed in the coexistence region.

Let us now show how our effective equilibrium theory allows us to account for the emerging behaviors of binary active mixtures when Eq. (3) is satisfied. For sake of generality, we allow both for global interactions, where the self-propulsion speed $v_\mu$ of species $\mu$ depends on the total density field of particles, $\rho_t(\mathbf{r}) = \sum_\mu \rho_\mu(\mathbf{r})$, and for specific ones, where $v_\mu$ depends specifically on one−or more−density field $\rho_v(\mathbf{r})$. In the latter case, we refer to self and cross interactions when $\mu = v$ and $\mu \neq v$, respectively.

We first consider self-regulation of motility by making $v_\mu$ depend on a local measurement of the density field of the same species, $\tilde{\rho}_\mu(\mathbf{r}) = K * \rho_\mu(\mathbf{r})$, where $K$ is a bell-shaped function and the asterisk indicates a convolution of this kernel with the density field. We thus write $v_\mu(\mathbf{r}) = \hat{v}_\mu^0 \phi_\mu^s[\tilde{\rho}_\mu(\mathbf{r})]$, where $\phi_\mu^s$ is an arbitrary function. In addition, we consider a second quorum-sensing circuit that makes the motility depend on the global density, so that $\hat{v}_\mu^0$ is itself a functional of $\rho_t$. All in all, the self-propulsion speed of species $\mu$ thus reads

$$v_\mu(\mathbf{r}) = v_\mu^0 \phi_\mu^s[\tilde{\rho}_\mu(\mathbf{r})]\phi_\mu^g[\tilde{\rho}_t(\mathbf{r})]. \quad (11)$$

A wealth of motility-control mechanisms can be described by adjusting the functions $\phi_\mu^s$ and $\phi_\mu^g$. Here we consider self-inhibition and global activation of motility by using decreasing and increasing sigmoidal functions for $\phi_\mu^s$ and $\phi_\mu^g$, respectively. This choice is motivated by synthetic-biology experiments where such orthogonal QS circuits can be engineered[68]. The non-local sampling of the densities through the convolutions by the kernel $K$ then results from a fast variable treatment on the signalling molecular field[69]. Alternatively, such interactions can be directly engineered for light-controlled active colloids[53,54]. To map out the phases accessible to the system, we carried out large-scale simulations of dynamics (1) using the QS interactions (11). Numerical details and the precise choices of $K$, $\phi_\mu^s$ and $\phi_\mu^g$ are described in Methods.

Varying the overall composition of the system reveals a rich phenomenology. First, as in single-species systems, the self-inhibition of a given species may lead to its motility-induced phase separation[70,71]. The second species then experiences a mild opposite modulation of its density field (Fig. 1a). Then, two phases specific to (active) mixtures emerge from the global coupling. First, Fig. 1b shows a segregated phase in which the two strains demix and undergo phase separation. Note that the global density field $\rho_t(\mathbf{r})$ remains homogeneous, contrary to what happens in the case of single-species phase separation shown in Fig. 1a. Second, Fig. 1c shows the existence of a triple-coexistence

regime leading to the joint observation of $\alpha$-rich, $\beta$-rich, and well-mixed phases, together with an overall phase-separation for $\rho_t(\mathbf{r})$. Let us now show how our effective equilibrium theory allows us to account for the phase diagram of the system quantitatively.

Under the local approximation $\tilde{\rho}_\mu(\mathbf{r}) \simeq \rho_\mu(\mathbf{r})$, Eq. (11) leads to $\log v_\mu = \log v_\mu^0 + \log \phi_\mu^s(\rho_\mu) + \log \phi_\mu^g(\rho_t)$. Direct inspection shows that Eq. (9) then amounts to $\phi_\mu^g = \phi_v^g$ for all $\mu, v$: the sole requirement for effective equilibrium is that the global interaction term $\phi_\mu^g$ be common to all species. This is the case for the system shown in Fig. 1a–c, which can thus be mapped onto an equilibrium problem. The self-organization of the two coexisting species can then be predicted from the analysis of the corresponding effective free energy, which we detail in the Methods. Departure from a homogeneous, well-mixed system will occur whenever the free energy density (10b) is not convex. Predicting the coexisting densities then amounts to constructing the tangent planes of $f(\rho_\alpha, \rho_\beta)$ (this is a standard equilibrium problem[72] that we detail in the Supplementary Information for completeness). In Fig. 2a, we compare these theoretical predictions to direct measurements for the parameters corresponding to Fig. 1a–c. Despite the construction relying on a locality asumption and a mean-field approximation, the agreement between predicted and measured phase diagrams in the composition space $(\rho_\alpha^0, \rho_\beta^0)$ is excellent. The triple coexistence regime reported in Fig. 1c emerges when the free energy surface admits a plane that is tangent in three points $(\rho_\alpha^i, \rho_\beta^i)_{i \in \{1,2,3\}}$ simultaneously, as illustrated in Fig. 2b. The corresponding compositions then delimit the coexistence region and determine the coexisting phases, while the respective fraction of each phase is obtained using the lever rule. Finally, the existence of an effective free energy also ensures that the Gibbs phase rule applies, which explains the existence of the three-phase and two-phase coexistence regions for our active binary mixture. The equilibrium mapping thus fully accounts for the static phase-separation scenario reported in Fig. 1. We now illustrate how violations of the microscopic condition (9) may lead to an emerging physics that explicitly breaks time-reversal symmetry.

## When non-reciprocal interactions survive coarse-graining

The violation of Eq. (3) is a sufficient condition for the emergence of non-reciprocal couplings between the density fields at the macroscopic scale (6). Consequently, the steady-state entropy production rate (7) is positive and the hydrodynamic equations do not have a gradient structure. In turn, it is well known that this allows for the

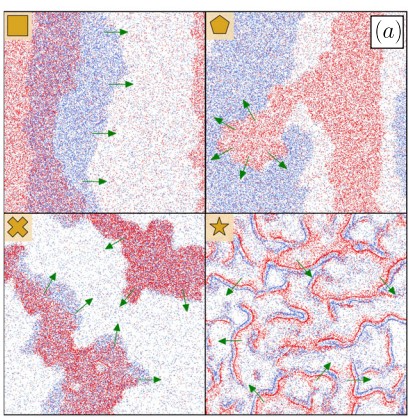
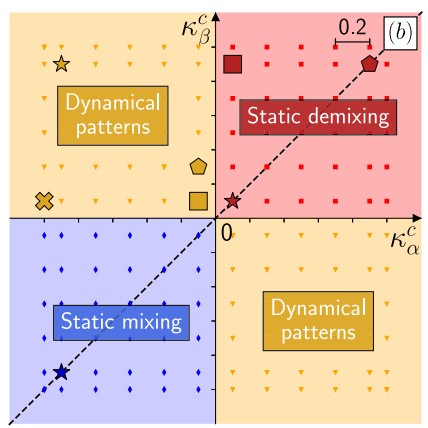
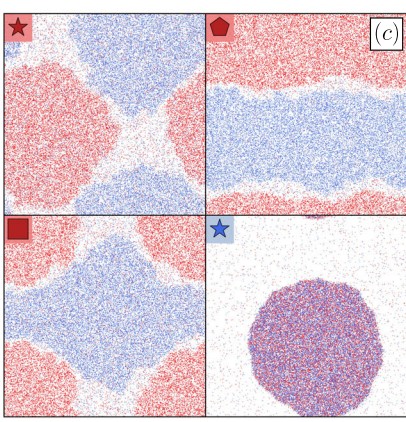

**Fig. 3 | Simulations of two species of RTPs when non-reciprocity survives coarse-graining.** Microscopic simulations of Eq. (15), with self-inhibition and non-reciprocal cross interactions. **a** and **c** Dynamical and static patterns observed in simulations, respectively; $\alpha$-particles are depicted in red, $\beta$-particles in blue. The snapshots correspond to the larger symbols shown in (**b**). **b** Phase diagram as the couplings $\kappa_\alpha^c$ and $\kappa_\beta^c$ are varied, where we remind that $\kappa_\mu^c > 0$ corresponds to the existence of travelling patterns, which can be studied using standard

activation of the motility of species $\mu$ by the density of species $\nu$, whereas $\kappa_\mu^c < 0$ corresponds to an inhibition. The background colors correspond to the predictions of linear stability analysis which are confirmed by numerical simulations (small symbols). The phase diagram is symmetric with respect to the dashed line $\kappa_\beta^c = \kappa_\alpha^c$ upon inverting the roles of $\alpha$- and $\beta$-particles in (**a**, **c**). See Methods for other parameters and numerical details.

tools[27–29] and whose irreversible dynamics contribute to the positive value of $\sigma$. The non-linearities of our field theory differ from recently proposed phenomenological descriptions of non-reciprocal scalar active systems[16,17,21–23] but their pattern-formation dynamics are expected to share similar features[27]. The main novelty, here, is that we can determine a *microscopic* condition for travelling patterns to emerge, since we are able to quantitatively connect microscopic and macroscopic scales. To do so, we consider the fate of perturbations around homogeneous solutions of Eq. (6), $\rho_\mu(\mathbf{r}) = \rho_\mu^0 + \delta\rho_\mu(\mathbf{r})$. In Fourier space, their linearized dynamics read $\partial_t\delta\tilde{\rho}_\mu(\mathbf{q}) = -\mathbf{q}^2\mathcal{M}_{\mu\nu}(\mathbf{q})\delta\tilde{\rho}_\nu$, with

$$\mathcal{M}_{\mu\nu}(\mathbf{q}) = D_\mu^0\left[\delta_{\mu\nu} + \rho_\mu^0\frac{\partial}{\partial\rho_\nu}\log v_\mu\right] \qquad (12)$$

where $D_\mu^0 \equiv D_\mu(\{\rho_\nu^0\})$. As shown in Methods, for $N = 2$ species, eigenvalues with non-vanishing imaginary parts require

$$\frac{\partial v_\beta}{\partial\rho_\alpha}\frac{\partial v_\alpha}{\partial\rho_\beta} < -\frac{v_\alpha v_\beta(\mathcal{M}_{\alpha\alpha} - \mathcal{M}_{\beta\beta})^2}{4D_\alpha^0 D_\beta^0 \rho_\alpha^0 \rho_\beta^0}. \qquad (13)$$

We thus predict an oscillatory behavior in the presence of sufficiently strong, opposite interactions, e.g., when species $\beta$ enhances the speed of $\alpha$ while $\alpha$ inhibits the motility of $\beta$. Under this condition, which is a stronger requirement than simple non-reciprocity, homogeneous profiles are linearly unstable whenever

$$\sum_{\mu=\alpha,\beta}D_\mu^0\left(1 + \rho_\mu^0\frac{\partial}{\partial\rho_\mu}\log v_\mu\right) < 0. \qquad (14)$$

This opens up the possibility of travelling patterns, which we now explore.

To do so, we consider a two-species system with self-inhibition of motility coupled to non-reciprocal cross-interactions. As for Eq. (11), we consider orthogonal QS circuits so that the self propulsion of species $\mu$ is given by

$$v_\mu(\mathbf{r}) = v_\mu^0\phi_\mu^s[\tilde{\rho}_\mu(\mathbf{r})]\phi_\mu^c[\tilde{\rho}_\nu(\mathbf{r})]. \qquad (15)$$

Self-inhibition is again modelled by using a decreasing sigmoidal function. To easily control the strength of the non-reciprocal

couplings, we choose $\phi_\mu^c(\tilde{\rho}) = \exp[\kappa_\mu^c\mathcal{S}_\mu^c(\tilde{\rho})]$, where $\mathcal{S}_\mu^c(\tilde{\rho})$ is a sigmoidal function described in Methods. Varying the values of $\kappa_\mu^c$ then allows us to model a variety of interactions: positive values of $\kappa_\mu^c$ correspond to motility enhancement while negative values lead to motility inhibition. In most of our simulations, we use self-inhibitions strong enough for Eq. (14) to hold, so that the system is never homogeneous. As shown in Fig. 3, our simulations reveal a variety of static and dynamical patterns (note that Eq. (7) predicts the entropy production rate $\sigma$ both in the presence of static and traveling patterns, irrespective of the nature of the steady state[73]). In agreement with our prediction (13), travelling patterns are observed when $\kappa_\alpha^c\kappa_\beta^c < 0$ (orange quadrants).

An intuitive microscopic understanding of the observed phases can be achieved by noticing that each species tends to accumulate where it goes slower[57,60,74–77]. Motility inhibition then acts as an effective attraction whereas motility enhancement leads to effective repulsion. When $\kappa_{\alpha,\beta}^c$ are both positive, the species effectively repel each other, leading to a triple coexistence regime with demixing between the dense phases (red quadrant). On the contrary, negative $\kappa_{\alpha,\beta}^c$ lead to effective attractive interactions and co-localization of the liquid phases (blue quadrant). The frustrated case, $\kappa_\alpha^c\kappa_\beta^c < 0$, corresponds to the motility of one species—say $\alpha$—being inhibited by the other—$\beta$ in this case—while that of $\beta$ is enhanced by $\alpha$. This leads to a complex run-and-chase dynamics between the two species that results in steady (orange square, Supplementary Movie 2) or chaotic (orange star, Supplementary Movie 3) travelling bands when $|\kappa_\alpha^c| \simeq |\kappa_\beta^c|$, as well as to a rich variety of more complex dynamical behaviors in less symmetric cases (orange cross and pentagon as well as Supplementary Movies 4–6). Thanks to the explicit coarse-graining of the microscopic dynamics, we are thus able to determine the microscopic condition for travelling patterns to emerge and to identify the mechanism leading to the run-and-chase dynamics.

## Non-reciprocity in chemotaxis
To show how our results generalize beyond the case of quorum sensing, we consider chemotactic interactions which have attracted a lot of interest in the context of bacterial suspensions[78–85]. For sake of generality, we consider $N$ species of ABPs/RTPs that evolve according to the dynamics (1) and whose motilities are biased by the gradients of $n$ chemical fields $\{c_p(\mathbf{r})\}$. We allow both for biases on the reorientation

dynamics and on the self-propulsion speeds:

$$v_\mu = v_{0\mu} - \mathbf{u}_{i,\mu} \cdot \sum_{p=1}^{n} v_{1\mu}^p \nabla_{\mathbf{r}_{i,\mu}} c_p, \qquad \tau_\mu^{-1} = \tau_{0\mu}^{-1} + \mathbf{u}_{i,\mu} \cdot \sum_{p=1}^{n} \left(\tau_{1\mu}^p\right)^{-1} \nabla_{\mathbf{r}_{i,\mu}} c_p,$$

(16)

where the parameters $v_{*\mu}, \tau_{*\mu}^{-1}$ are constant. When $v_{1\mu}^p, \tau_{1\mu}^p$ are positive, particles increase their persistence lengths when moving toward minima of $c_p$, implying that $c_p$ acts as a chemorepellent. Conversely, negative values of $v_{1\mu}^p, \tau_{1\mu}^p$ correspond to chemoattraction. We consider the case in which the chemicals are produced by the particles before they diffuse and degrade in the environment at non-vanishing rates. In the large system-size limit, the dynamics of $c_p$ is thus much faster than that of the conserved density field $\rho_\mu$ and the chemical fields follow the evolution of the density fields adiabatically: $c_p(\mathbf{r}) \equiv c_p(\mathbf{r}, [\{\rho_\nu\}])$.

We start by coarse-graining the microscopic dynamics into a stochastic field theory for the density fields, which takes the form of Eq. (6) with an effective chemical potential given by (see Supplementary Information):

$$u_\mu = \frac{1}{v_{0\mu}^2} \sum_{p=1}^{n} \left(\frac{v_{1\mu}^p}{\tau_{0\mu}} + \frac{v_{0\mu}}{\tau_{1\mu}^p}\right) c_p + \log \rho_\mu.$$

(17)

Consequently, the entropy production rate remains given by Eq. (7) albeit with $u$ determined by Eq. (17). The integrability condition for non-reciprocity to vanish across scales is then given by the functional Schwarz theorem as:

$$\forall(\mu, \nu), \qquad \frac{1}{v_{0\mu}^2} \sum_{p=1}^{n} \left(\frac{v_{1\mu}^p}{\tau_{0\mu}} + \frac{v_{0\mu}}{\tau_{1\mu}^p}\right) \frac{\delta c_p(\mathbf{r})}{\delta \rho_\nu(\mathbf{r}')} = \frac{1}{v_{0\nu}^2} \sum_{p=1}^{n} \left(\frac{v_{1\nu}^p}{\tau_{0\nu}} + \frac{v_{0\nu}}{\tau_{1\nu}^p}\right) \frac{\delta c_p(\mathbf{r}')}{\delta \rho_\mu(\mathbf{r})}.$$

(18)

Equation (18) determines when a microscopic chemotactic dynamics admits a large-scale effective equilibrium description.

For sake of concreteness, let us consider the simplest case of a single chemical field ($n = 1$) given by:

$$c(\mathbf{r}, [\{\rho_\nu\}]) = \sum_\mu \beta_\mu \tilde{\rho}_\mu(\mathbf{r}),$$

(19)

where $\tilde{\rho}_\mu = K * \rho_\mu, \beta_\mu$ is the production rate of $c$ by species $\mu$, and $K$ is the Green's function corresponding to the linear transport and degradation of the chemicals. For the sake of simplicity, we only consider biases on the self-propulsion speeds and set $v_{0\mu} \equiv v_0, \tau_{0\mu} \equiv \tau_0$ and $\tau_{1\mu}^{-1} = 0$ for all species. In the particle dynamics, chemotactic interactions can then be seen as "generalized" pairwise forces:

$$\dot{\mathbf{r}}_{i,\mu} = v_0 \mathbf{u}_{i,\mu} + \sum_{j,\nu} \mathbf{f}_{i,\mu}^{j,\nu}, \quad \text{where} \quad \mathbf{f}_{i,\mu}^{j,\nu} = v_{1\mu} \beta_\nu \mathbf{u}_{i,\mu} [\mathbf{u}_{i,\mu} \cdot \nabla_{\mathbf{r}_{i,\mu}} K(\mathbf{r}_{i,\mu} - \mathbf{r}_{j,\nu})].$$

(20)

As for the QS interactions described in Eq. (2), the dynamics is clearly non-reciprocal at the microscopic scale. Indeed, $\mathbf{f}_{i,\mu}^{j,\nu}$ and $\mathbf{f}_{j,\nu}^{i,\mu}$ are not collinear, since $\mathbf{f}_{i,\mu}^{j,\nu}$ is directed along $\mathbf{u}_{i,\mu}$, and Newton's third law is violated. Nevertheless, Eq. (18) ensures that reciprocity is restored at the coarse-grained scale whenever $v_{1\mu}\beta_\nu = v_{1\nu}\beta_\mu$ for all species. We note that equilibrium limits of chemotactic[86,87] or diffusiophoretic dynamics[12,15,20] have attracted a long-standing interest in the literature. Previous results, however, relied on microscopic Langevin dynamics in which chemotactic interactions enter directly as effective pairwise collinear forces, $\mathbf{f}_i^j \propto \nabla_i G(\mathbf{r}_i - \mathbf{r}_j)$ for some function $G$. The existence of

macroscopic equilibrium limits then relies on imposing Newton's third law at the microscopic scale. On the contrary, Eq. (18) is, to the best of our knowledge, the first condition for chemotactic mixtures to recover reciprocity at the macroscopic scale, despite being non-reciprocal at the microscopic one.

## Discussion

In this article we have shown how microscopic and macroscopic scales can be quantitatively bridged for a large class of active mixtures in the presence of mediated non-reciprocal interactions. This revealed a subtle and important property of non-reciprocity: it varies strongly across scales. Based on this insight we derived non-trivial conditions on the *microscopic* NRI that lead to effective equilibrium at the *macroscopic* scale. This allowed us to account—accurately and without fit parameters—for the full range of static patterns observed in our simulations. Finally, we derived conditions for NRI to survive coarse-graining, hence leading to positive entropy production rate at the macroscopic scale. When non-reciprocity is strong enough, we showed the emergence of a wealth of dynamical patterns. Again, our micro-to-macro approach allows us to predict the phase diagram from microscopics without fitting parameters.

From a biophysical perspective, our study shows how QS and chemotactic interactions lead to a rich phenomenology in complex assemblies of cells. In the context of bacterial colonies, motility-induced patterns will eventually interact with population dynamics[68,88,89] and genetics[90]. How this interplay will result in diverse co-existing communities is a fascinating research direction for the future indeed.

Next, we note that swimming bacteria like E. coli typically grow up to 0.1% volume fraction[91], so that steric interactions can safely be neglected. It is however possible to design experiments in which bacterial density is much larger, e.g., in swarming conditions[92]. A natural question is then how our results extend to such systems. As shown in the Supplementary Information and in Supplementary Movie 7, a large part of the phenomenology discussed in this article can be found in mixtures of active particles interacting both via pairwise repulsive forces and QS interactions. Recent theoretical progress places the analytical description of this case within the reach of future work[35,93–100].

Finally, turning synthetic active-matter systems into smart materials will require quantitative control over complex assemblies of active constituents. Our work demonstrates that one can go up the complexity ladder while retaining an analytical framework to account for the emerging properties of active systems. How these systems can then be optimized to accomplish given tasks is an exciting challenge that appears within reach, given recent progress in automatic differentiation[101].

## Methods
### Simulations
All our numerics were carried out using run-and-tumble dynamics in continuous space, with discrete time-steps and periodic boundary conditions. For a particle of species $\mu$, the tumbles are implemented as follows: at $t = 0$ or after a tumble at time $t$, the time until the next tumble $\delta t$ is sampled from an exponential distribution: $p(\delta t) = \exp(-\delta t/\tau_\mu)/\tau_\mu$. The particle then moves in a straight line until $t + \delta t$. Since we use discrete time steps, there exists a time step such that $n\, dt < t + \delta t < (n+1) dt$. During this time step, the particle first moves with the velocity evaluated at $t = n\, dt$, until $t + \delta t$. A new direction is then chosen uniformly at random, the next tumbling time is sampled from $p(\delta t)$, and the particle resumes the time step with the same speed and a new orientation, until either $t + dt$ or the next tumble occurs. (Multiple tumbles can occur in a single time step.)

To compute the self-propulsion speed of particle $i$ of species $\mu$, we first determine $\tilde{\rho}_\nu(\mathbf{r}_{i,\mu})$ through:

$$\tilde{\rho}_\nu(\mathbf{r}_{i,\mu}) = \int \mathrm{d}^2\mathbf{z}\, K(\mathbf{r}_{i,\mu} - \mathbf{z})\rho_\nu(\mathbf{z})\,,$$

$$\text{with}\quad K(\mathbf{r}) = \frac{1}{Z}\exp\left(-\frac{r_{QS}^2}{r_{QS}^2 - \mathbf{r}^2}\right)\Theta\left(r_{QS}^2 - \mathbf{r}^2\right), \tag{21}$$

where $\Theta$ is the Heaviside function. In all simulations, we set the unit of length such that $r_0 = 1$. Since $\rho_\nu(\mathbf{z}) = \sum_{j=1}^{N_\nu}\delta(\mathbf{z} - \mathbf{r}_{j,\nu})$, the computation of $\tilde{\rho}_\nu(\mathbf{r}_{i,\mu}) = \sum_{j=1}^{N_\nu}K(\mathbf{r}_{i,\mu} - \mathbf{r}_{j,\nu})$ appears to scale as $\mathcal{O}(N_\nu)$. However, it can be implemented in $\mathcal{O}(1)$ operations since $K$ has a compact support. To do so, we use a spatial hashing and divide the total space into cells of linear size $r_{QS}$. Then, we only need to consider the particles in the 9 boxes nearest to $\mathbf{r}_{i,\mu}$ to compute $\tilde{\rho}_\nu(\mathbf{r}_{i,\mu})$.

Once the values of $\tilde{\rho}_\nu(\mathbf{r}_{i,\mu})$ have been evaluated for all $(i, \mu, \nu)$, we determine the particle speeds. For the reciprocal interactions, we follow Eq. (11) and use $v_\mu(\mathbf{r}) = v_\mu^0 \phi_\mu^s[\tilde{\rho}_\mu(\mathbf{r})]\phi_\mu^g[\tilde{\rho}_t(\mathbf{r})]$, where the sigmoidal functions $\phi_\mu^g$ and $\phi_\mu^s$ are given by

$$\phi_\mu^s(\tilde{\rho}_\mu) \equiv \exp\left[\kappa_\mu^s \tanh\left(\frac{\tilde{\rho}_\mu - \bar{\rho}}{\delta\rho}\right)\right] \quad\text{and}\quad \phi_\mu^g(\tilde{\rho}_t) \equiv \exp\left[\kappa_\mu^g \tanh\left(\frac{\tilde{\rho}_t - 2\bar{\rho}}{2\delta\rho}\right)\right]. \tag{22}$$

Our specific choices for the functional forms of $\phi_\mu^{s,g}$ proves useful below to make progress analytically, but any other sigmoidal functions would lead to qualitatively similar results. For the non-reciprocal interactions, we follow Eq. (15) and use $v_\mu(\mathbf{r}) = v_\mu^0 \phi_\mu^s[\tilde{\rho}_\mu(\mathbf{r})]\phi_\mu^c[\tilde{\rho}_\nu(\mathbf{r})]$, where $\phi_\mu^s$ is still given by Eq. (22) and

$$\phi_\mu^c[\tilde{\rho}_\nu] \equiv \exp\left[\kappa_\nu^c \tanh\left(\frac{\tilde{\rho}_\nu - \bar{\rho}}{\delta\rho}\right)\right]. \tag{23}$$

Finally, for all simulations, we use adaptive time-stepping such that the maximal displacement $\max(v_{i,\mu}dt)$ is always smaller than $r_{QS}/5$. The time steps indicated below are thus the largest time steps used in the simulations.

**Effective free energy**

When the entropy production rate given by Eq. (7) vanishes, the macroscopic dynamics amounts to a bona-fide equilibrium one. In this section, we derive the explicit form of the corresponding free energy when, as in our simulations, the self-propulsion speed is given by Eq. (22).

In the local approximation, the condition (9) of the main text applied to the self-propulsion speed (11) reduces to $\phi_\mu^g(\rho_t) \equiv \phi^g(\rho_t)$ for all species. For our specific choice of $v_\mu$, this simply translates into $\kappa_\alpha^g = \kappa_\beta^g \equiv \kappa^g$. When this condition holds, we can integrate the chemical potential $\mathrm{u}_\mu = \log v_\mu + \log \rho_\mu$ with respect to $\rho_\mu$ to obtain the free energy density:

$$f(\rho_\alpha, \rho_\beta) = \underbrace{2\kappa^g \delta\rho \log\left[\cosh\left(\frac{\rho_t - 2\bar{\rho}}{2\delta\rho}\right)\right] + \sum_{\mu=\alpha,\beta}\kappa_\mu^s \delta\rho \log\left[\cosh\left(\frac{\rho_\mu - \bar{\rho}}{\delta\rho}\right)\right]}_{\equiv U(\rho_\alpha, \rho_\beta)}$$
$$+ \underbrace{\sum_{\mu=\alpha,\beta}\rho_\mu \log \rho_\mu}_{\equiv -s(\rho_\alpha, \rho_\beta)} \tag{24}$$

where $\rho_t = \rho_\alpha + \rho_\beta$. Eq. (24) can be directly interpreted as an interplay between an effective energy density $U$ and an entropy density $s$. Note, however, that $U$ has no microscopic interpretation in terms of an energy function.

Once the explicit form of $f$ is known, we can construct the associated phase diagram via a common-tangent construction, exactly as for an equilibrium system. For completeness, the corresponding

procedure is detailed in the Supplementary Information and we refer the interested reader to standard reviews for further details[72].

**Figure parameters**

- Fig. 1: system size $150 \times 150$, $dt = 0.004$, $\bar{\rho} = 50$, $\delta\rho = 20$, $v_{\alpha,\beta}^0 = 5$, $\tau_{\alpha,\beta} = 1$, $r_{QS} = 1$, strength of self-inhibition $\kappa_{\alpha,\beta}^s = -0.8$, strength of activation $\kappa_{\alpha,\beta}^g = 0.8$. Simulations are initialized in a homogeneous configuration with densities $\rho_\alpha^0, \rho_\beta^0$. The species compositions vary between the three panels: $\rho_\alpha^0 = 15, \rho_\beta^0 = 50$ in panel (a); $\rho_{\alpha,\beta}^0 = 55$ in panel (b); $\rho_{\alpha,\beta}^0 = 75$ in panel (c).
- Fig. 2: same parameters as in Fig. 1.
- Fig. 3: system size $30 \times 30$, $dt = 0.005$, $\bar{\rho} = 25$, $\delta\rho = 10$, $v_{\alpha,\beta}^0 = 5$, $\tau_{\alpha,\beta} = 1$, $r_{QS} = 1$, strength of self-inhibition $\kappa_{\alpha,\beta}^s = -1$, $\rho_{\alpha,\beta}^0 = 25$.

**Linear stability analysis**

Here we derive the linear dynamics of small perturbations $\delta\rho_\mu(\mathbf{r})$ around homogeneous density profiles $\rho_\mu^0$. Our starting point is the mean-field approximation of the fluctuating hydrodynamics (6). As in our simulations, we consider constant tumbling rates, and self-propulsion speeds $v_\mu$ of the form:

$$v_\mu(\mathbf{r}, [\rho_1, \ldots, \rho_N]) = v_\mu(\tilde{\rho}_1(\mathbf{r}), \ldots, \tilde{\rho}_N(\mathbf{r})) \tag{25}$$

where $\tilde{\rho}_\mu$ corresponds to the coarse-grained measurement of the local density field $\rho_\mu$ through Eq (21). Linearization of the mean-field hydrodynamics then leads to:

$$\delta\dot{\rho}_\mu = D_\mu^0\left[\rho_\mu^0 \sum_{\nu=1}^N \partial_\nu \log v_\mu \nabla^2(K * \delta\rho_\nu) + \nabla^2 \delta\rho_\mu\right] \tag{26}$$

where $D_\mu^0 \equiv D_\mu(\{\rho_\nu^0\})$ and $\partial_\nu \log v_\mu \equiv \left.\frac{\partial}{\partial\rho_\nu}\log v_\mu\right|_{\rho^0}$. In Fourier space, Eq. (26) becomes $\partial_t \delta\hat{\rho}_\mu(\mathbf{q}) = -\mathbf{q}^2 \mathcal{M}_{\mu\nu}(\mathbf{q})\delta\hat{\rho}_\nu(\mathbf{q})$, where the expression of the matrix $\mathcal{M}_{\mu\nu}$ is given in Eq. (12) of the main text. The diagonal elements of $\mathcal{M}$ contain the self-interaction terms $\partial_\mu \log v_\mu$, while off-diagonal terms depend on cross-interactions $\partial_\nu \log v_\mu$. So far, our linear stability analysis retains the full non-locality due to the kernel $K$. To consider a local or quasi-local approximation, we note that a gradient expansion of $\rho(\mathbf{r} - \mathbf{r}')$ to second order yields

$$\tilde{\rho}(\mathbf{r}) \approx \rho(\mathbf{r}) + \frac{1}{2}\gamma^2 \Delta\rho(\mathbf{r}), \qquad \gamma^2 = \int \mathrm{d}^d\mathbf{r}'K(\mathbf{r}')(\mathbf{r}')^2 \tag{27}$$

where we have exploited the normalization, symmetry, and isotropy of the kernel $K$. In the linear stability matrix given by Eq. (12), this amounts to replacing

$$\hat{K}(\mathbf{q}) \qquad \text{by} \qquad 1 - \frac{1}{2}\gamma^2 q^2 + \mathcal{O}(q^4). \tag{28}$$

Taking $\gamma = 0$ leads to the purely local approximation discussed in the main text.

**Traveling waves in binary mixtures.** In the $N = 2$ case the dynamical matrix $\mathcal{M}_\mathbf{q}$ takes the form:

$$\mathcal{M}(\mathbf{q}) = q^2 \begin{pmatrix} D_\alpha^0\left(1 + \hat{K}(\mathbf{q})\rho_\alpha^0 \partial_\alpha \log v_\alpha\right) & D_\alpha^0 \rho_\alpha^0 \hat{K}(\mathbf{q})\partial_\beta \log v_\alpha \\ D_\beta^0 \rho_\beta^0 \hat{K}(\mathbf{q})\partial_\alpha \log v_\beta & D_\beta^0\left(1 + \hat{K}(\mathbf{q})\rho_\beta^0 \partial_\beta \log v_\beta\right) \end{pmatrix} \tag{29}$$

and the associated eigenvalues can be computed as:

$$\lambda(\mathbf{q}) = \frac{\text{Tr}\,[\mathcal{M}(\mathbf{q})] \pm \sqrt{\Delta[\mathcal{M}(\mathbf{q})]}}{2} \quad \text{with} \quad \Delta[\mathcal{M}(\mathbf{q})] = \text{Tr}\,[\mathcal{M}(\mathbf{q})]^2 - 4\,\text{Det}\,[\mathcal{M}(\mathbf{q})].$$
(30)

A traveling perturbation of the homogeneous profile arises at linear order whenever $\text{Im}[\lambda] \neq 0$. This is equivalent to $\Delta[\mathcal{M}] < 0$, i.e.,:

$$\left(\mathcal{M}_{\alpha\alpha} - \mathcal{M}_{\beta\beta}\right)^2 + 4\hat{K}(\mathbf{q})^2 D_\alpha^0 D_\beta^0 \rho_\alpha^0 \rho_\beta^0 (\partial_\beta \log v_\alpha)(\partial_\alpha \log v_\beta) < 0, \quad (31)$$

which corresponds to Eq. (13) of the main text in the local approximation $K(\mathbf{q}) = 1$. We note that Eq. (31) requires

$$\partial_\beta v_\alpha \cdot \partial_\alpha v_\beta < 0, \quad (32)$$

and thus that cross-interactions act in opposite ways. We stress that this result holds at the fully non-local level and not solely in the local approximation discussed in the main text.

If $\Delta[\mathcal{M}] < 0$, the growth rate of the perturbation is $-\text{Tr}\,[\mathcal{M}]/2$. For a homogeneous phase to be linearly unstable, one thus needs $\text{Tr}\,[\mathcal{M}] < 0$; this condition leads, in the local case, to Eq. (14) of the main text.

As a final remark we note that, much like phase separation can be observed outside the spinodal region, traveling states can be observed even when linear stability analysis would predict a homogeneous phase to be linearly stable, as shown in Fig. S2 of the Supplementary Information.

## Data availability
The data that support the findings of this study are available from the corresponding author upon request. The source data generated in this study have been deposited in the Figshare database under accession code https://doi.org/10.6084/m9.figshare.23713455.

## Code availability
The codes and algorithms that have been used to support the findings of this study are available from the corresponding author upon request.

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

## Acknowledgements

J.T. acknowledges the financial support of ANR Thema. A.D. acknowledges an international fellowship from Idex Universite de Paris. P.S. acknowledges support by a RSE Saltire Facilitation Network Award. Y.Z. acknowledges support from start-up grant NH10800621 from Soochow University.

## Author contributions

A.D., J.O., A.C., Y.Z., P.S., and J.T. conceived the project and wrote the paper. A.D. produced all numerical data. A.D., J.O., and A.C. carried out the analytical study.

## Competing interests

The authors declare no competing interests.
