## [Peer Review File · Nature Communications]

REVIEWER COMMENTS

Reviewer #1 (Remarks to the Author):

The authors study the collective properties of a mixture of active particles interacting via non-reciprocal interactions. As a starting point, they consider a mixture of self-propelled particles interacting through quorum sensing interactions. In quorum sensing, the self-propelled velocity depends on the local density so that, once one accounts for mixtures of different species, the interactions result be non-reciprocal by definition. Using standard manipulations, they develop the large-scale description of the system and test the prediction of the theory against numerical simulations of the microscopic dynamics in the case of a non-reciprocal binary mixture. They show that in some situations non-reciprocal interactions do not survive the coarse-graining procedure meaning that the system arranges in phase-separated phases that can be rationalized in terms of an effective free energy functional. In other circumstances, i.e., when the degree of non-reciprocity is strong enough, non-reciprocal interactions might cause dynamical patterns as traveling waves. Finally, they explore the impact of non-reciprocal interactions in the case of chemotaxis.

I think the results might be of interest to the reader of Nature Communications. I support the publication of the manuscript on the condition they address my comments listed below.

1. While it is clear that, because of the breaking of Newton's third law, momentum is not conserved the other way around does not sound totally obvious. In particular, in the abstract, they write "the lack of momentum makes non-reciprocal interactions the rule rather than the exception" that sounds quite strong to me. About that, I think the authors should take into account "Statistical Mechanics where Newton's Third Law is Broken" PRX 5, 011035 (2015) where it is shown how in presence of the breaking of Newton's third law a pseudomomentum and pseudoenergy conservation might still happen.

2. In the end the mechanism leading to traveling waves might sound exactly the same as discussed in Ref. [17], I would appreciate it if the authors might clarify the novelty of their approach within the framework of scalar active mixtures. I think the mechanism leading to traveling waves in the present work is directly linked to non-equilibrium on the macroscopic scale signaled by a non-vanishing entropy production rate. Is not it?

Minor comments

1. I would suggest to remove reference to work in preparations that are not published/on arxiv
2. What is the message they want to deliver by introducing the Lyapunov functional?

Reviewer #2 (Remarks to the Author):

Starting from mixtures of active particles that experience very general non-reciprocal quorum sensing interactions, the authors develop a coarse graining approach following standard procedures. Based on a vanishing entropy rate, the authors show that the coarse-grained dynamics can be derived from an effective free energy, which means its dynamics is reciprocal. They exemplify the resulting dynamics in both cases and also comment on chemotactic interactions.

The submitted manuscript certainly presents well researched work. However, I cannot recommend it for Nature Communications. An article in this journal should be presented without that one needs to constantly rely on the supplement material. The manuscript is written for insiders and stays mostly on an abstract level, without making a real effort to clearly explain what the conditions for reciprocity or non-reciprocity in the microscopic equations mean. I also think that the quorum sensing interaction describes a very special interaction. For example, denser bacterial systems would immediately need steric interactions between rods etc. So one can strongly question the significance of this work for the field.

I have a few more comments:

1. Eq. (6) results from a definition of the entropy production rate, which should certainly appear in the main text since σ is the crucial quantity here. Eq. (6) uses steady-state distributions. Does this mean all what the authors derive here is only valid in steady state?
2. Certainly more information on Eq. (10) is needed. Why is it constructed like this.
3. Fig. 1 needs to be better discussed/explained.
4. Again: Eq. (14) needs more explanation.

Reviewer #3 (Remarks to the Author):

The authors study the collective behavior of self-propelled particles with non-reciprocal quorum-sensing interactions. They derive the fluctuating hydrodynamics of their microscopic model [eq.(3)], which allows them to provide a series of predictions. First, they obtain an explicit condition for an equilibrium mapping (EM) to exist at hydro level [eq.(7)], associated with vanishing entropy production rate. Within this mapping, they predict the phase diagram, and confirm its validity with particle-based simulations [figs.1-2]. Second, when the EM does not hold, they provide an explicit criterion for travelling patterns to appear [eq.(12)], and confirm again its validity with particle-based simulations [fig.3]. Finally, they consider non-reciprocity in chemotaxis, and show how the condition for EM translates in this context [eq.(17)]

The manuscript is well written, and provides a very nice addition to the existing literature on non-reciprocal interactions for active mixtures. Importantly, the EM provides an explicit constraint for when microscopic non-reciprocity becomes irrelevant at hydrodynamic level. This constraint clearly demonstrates how coarse-graining methods can delineate when the minimization of an effective free energy is indeed legitimate to predict instabilities. Moreover, the agreement between predictions and simulations is indeed excellent, for all cases (even beyond the regime of effective free energy), without any fitting parameters.

Therefore, I recommend publication of the manuscript in Nature Communications. Please find below some minor comments that the authors might want to consider.

1. In the introduction, the authors justify non-reciprocity in active systems by arguing that they 'exchange momentum with their environment'. It is unclear why this condition should actually open the door to non-reciprocal interactions. Maybe, the authors could clarify their statement by arguing why, in contrast, passive systems do not typically 'exchange momentum with their environment' (if they do not).

2. In the introduction, it could be worth providing explicit references for each of the three types of self propulsion considered here: active Brownian particles, run-and-tumble particles, and active Ornstein-Uhlenbeck particles. Likewise, when mentioning the entropy production rate (and also

around eq.(6)), it could be useful to provide more context and references: How is it defined? How is it connected to heat and irreversibility?

3. In the conclusion, the authors could mention other attempts to coarse-grain the dynamics, and to find an equilibrium mapping, for self-propelled particles beyond the case of quorum-sensing interactions. Indeed, for pair-wise interactions, analyzing the dynamics of the density field can prove more challenging. Here are recent attempts: J Stat Mech 2017, 113208; New J Phys 23, 103024 (2021); arXiv:2301.12155.

Dear Editor, dear Referees

We thank the three referees for taking the time to review and assess our manuscript. Referee 1 and 3 recommend publication of our article and make a number of comments that we have all addressed.

Referee 2 questions the relevance of our work for Nature Communications for two reasons: 1) the readability would be limited due to many references to the supplementary; 2) the scope of our results would be limited because restricted to quorum-sensing interactions.

Regarding point 1), we had tried to follow Nature Communications' guidelines and we can only apologize if this made the reading of the manuscript sometimes difficult. We have now tried to give all information required to understand the article in the main text. The supplementary thus contains only detailed derivations, which should be of interest only for specialists. We note that, because our results are general, we have separated their presentations from the numerical details of the examples on which they are illustrated. We think that this is the best way to show the generality of our approach. Details of simulations are thus left to the Methods section, as we believe is customary for Nature Communications' articles.

Regarding point 2), we first stress that our results were also derived for chemotaxis, and not solely for quorum sensing. Taken together, quorum sensing and chemotaxis are by far the most studied interactions for bacterial systems. Indeed, in nature, swimming bacteria grow up to typical densities that are way below 1% volume fraction so that pairwise forces, despite their appeal to physicists, have attracted less attention. Nevertheless, we agree with the referee that steric interactions are important, for instance for swarming bacteria. We have thus complemented our work by the study of a system in which particles interact both with reciprocal pairwise forces and with nonreciprocal quorum-sensing interactions. (Nonreciprocal pairwise forces that are not mediated are hard to imagine as being relevant for bacteria.) We show in the new SI Movie 6 that the phenomenology predicted for QS interactions in Figs. 1, 2 and 3 can also be observed in such a system.

We hope that the modifications we have made to the presentation of our work, together with the above discussion and the new example will convince the referee of the breadth of our results.

We provide below a detailed discussion of all reports, in which the referees' comments are italicized. We are painfully aware of the time it takes to review articles and we have thus also appended the results of the 'latexdiff' program that we ran on the manuscript and supplementary to produce comparisons between the previously submitted versions and the resubmitted ones. New additions appear in blue while deletions are in red. This makes this rebuttal very long, but we hope this will save the referees some time.

Report of reviewer #1

Summary

The authors study the collective properties of a mixture of active particles interacting via non-reciprocal interactions. As a starting point, they consider a mixture of self-propelled particles interacting through quorum sensing interactions. In quorum sensing, the self-propelled velocity depends on the local density so that, once one accounts for mixtures of different species, the interactions result be non-reciprocal by definition. Using standard manipulations, they develop the large-scale description of the system and test the prediction of the theory against numerical simulations of the microscopic dynamics in the case of a non-reciprocal binary mixture. They show that in some situations non-reciprocal interactions do not survive the coarse-graining procedure meaning that the system arranges in phase-separated phases that can be rationalized in terms of an effective free energy functional. In other circumstances, i.e., when the degree of non-reciprocity is strong enough, non-reciprocal interactions might cause dynamical patterns as traveling waves. Finally, they explore the impact of non-reciprocal interactions in the case of chemotaxis.

Review.

I think the results might be of interest to the reader of Nature Communications. I support the publication of the manuscript on the condition they address my comments listed below.

Response: We thank the referee for this precise and positive assessment of our work and for the comments below, which have helped us improve our manuscript.

1. While it is clear that, because of the breaking of Newton's third law, momentum is not conserved the other way around does not sound totally obvious. In particular, in the abstract, they write "the lack of momentum makes non-reciprocal interactions the rule rather than the exception" that sounds quite strong to me.

Response: We apologize for the confusion induced by our somewhat useless discussion of momentum conservation. What we meant was that the presence of momentum reservoirs, that are required for activity, also make mediated interactions the rule, rather than the exception. In turn, mediated interactions need not be constrained by Newton's third law, so that we expect non-reciprocal interactions to be generic. We now write "Mediated interactions, which are not constrained by Newton's third law, make non-reciprocal interactions a generic feature of active systems ". We hope that this statement will be clearer to the audience.

About that, I think the authors should take into account "Statistical Mechanics where Newton's Third Law is Broken" PRX 5, 011035 (2015) where it is shown how in presence of the breaking of Newton's third law a pseudomomentum and pseudoenergy conservation might still happen.

Response: We thank the referee for pointing out this reference, which we now cite twice: when we first define non-reciprocal forces and in a footnote one line below, which states "Note that, for some passive systems, it has been shown that non-reciprocal interactions may allow for a conservation law of a generalized momentum-like quantity [26]."

Referee's comment : *2. In the end the mechanism leading to traveling waves might sound exactly the same as discussed in Ref. [17], I would appreciate it if the authors might clarify the novelty of their approach within the framework of scalar active mixtures. I think the mechanism leading to traveling waves in the present work is directly linked to non-equilibrium on the macroscopic scale signaled by a non-vanishing entropy production rate. Is not it?*

Response: The referee is right that a non-vanishing entropy production rate is necessary to have a macroscopic dynamics that violates time-reversal symmetry and to observe dynamical patterns. This is true here and in all non-linear dynamics leading to traveling patterns. We now state this in a clearer manner at the beginning of Section 4.

Regarding the differences with recently proposed phenomenological descriptions of non-reciprocal scalar active systems [16,17,21-23], we believe that all share similarities. What singles out our work from all others is the direct connection to microscopic parameters that allows us to determine a microscopic condition for travelling patterns to emerge. (The non-linearities of our field theory also differ from those of existing works, but this is less important.) To clarify this, we now write:

“The non-linearities of our field theory differ from recently proposed phenomenological descriptions of non-reciprocal scalar active systems [16,17,21-23] but their pattern-formation dynamics are expected to share similar features [73]. The main novelty, here, is that we can determine a *microscopic* condition for travelling patterns to emerge, since we are able to quantitatively connect microscopic and macroscopic scales.”

Minor comments

1. *I would suggest to remove reference to work in preparations that are not published/on arxiv*

Response: We have now done so in the main text and in the supplementary.

2. *What is the message they want to deliver by introducing the Lyapunov functional?*

Response: We simply wanted to state that the equilibrium mapping does not simply apply to the steady state but also to the relaxational dynamics. We have now clarified this by writing: “Furthermore, \mathcal{F} is mathematically equivalent to a Landau free energy in equilibrium and plays the role of a Lyapunov functional for the dynamics, since $\partial_t \langle \mathcal{F} \rangle = - \int d\mathbf{r} \sum_{\mu} \langle M_{\mu} (\nabla \frac{\delta \mathcal{F}}{\delta \rho_{\mu}})^2 \rangle < 0$. This shows that the dynamics relaxes towards density profiles that minimize the Landau free energy \mathcal{F} , hence allowing us to determine the most probable configurations of the system using a variational principle.”

Report of reviewer #2

Summary

Starting from mixtures of active particles that experience very general non-reciprocal quorum sensing interactions, the authors develop a coarse graining approach following standard procedures. Based on a vanishing entropy rate, the authors show that the coarse-grained dynamics can be derived from an effective free energy, which means its dynamics is reciprocal. They exemplify the resulting dynamics in both cases and also comment on chemotactic interactions.

Review

The submitted manuscript certainly presents well researched work. However, I cannot recommend it for Nature Communications. An article in this journal should be presented without that one needs to constantly rely on the supplement material.

Response: We thank the referee for raising this point. In the resubmission, we have made sure that all important information (like the definition of the entropy production rate) is in the main text. Note that the Supplementary Information contains:

1. Algebra (the derivation of the coarse-grained dynamics in section 1, the computation of the entropy production rate in section 3)
2. The proof of the functional Schwarz theorem in section 2
3. The common tangent construction in section 4
4. The new case involving pairwise forces that you suggested in section 5

We stress that items 1 and 2 are not required to understand our results, but are simply there to help other scientists reproduce them. Item 3 is a textbook problem which is included for the sake of completeness. The sole part that is not described in depth in the main text is then the new addition suggested by the referee (the new section 5). We thus do not think that frequent interruptions to read the supplemental material will be needed to understand the article.

Regarding the references to the ‘Methods’ part, which may also have disturbed the referee, we have followed the recommendation of Nature Communication to separate the main results from the detailed methods. In the ‘Methods’ part, we now have

1. Simulation details.
2. Expression of the free energy for the simulated case.
3. Figure parameters.
4. The linear stability analysis.

Strictly speaking, none of these items are needed to understand the main text but they are needed to reproduce our results. We believe that the generality of our approach is an important feature of our work. We thus wanted to keep as separate as possible our general results from the numerical examples that we use for illustration. Furthermore, apart from figure captions, there are only four references to the methods, which should not hinder too strongly the reading of the article.

The manuscript is written for insiders and stays mostly on an abstract level, without making a real effort to clearly explain what the conditions for reciprocity or non-reciprocity in the microscopic equations mean.

Response: We thank the referee for raising this point, which has helped us clarify the definition of non-reciprocity in the microscopic model. We now explicitly address this question when discussing the quorum-sensing dynamics, in Eq. (2), so that non-reciprocity is clearly defined both for quorum-sensing and chemotactic interactions (in Eq. (20), which was already part of the initial submission).

I also think that the quorum sensing interaction describes a very special interaction. For example, denser bacterial systems would immediately need steric interactions between rods etc. So one can strongly question the significance of this work for the field.

Response: This is an interesting point. When grown in liquid environment, *E. coli* bacteria typically reach a volume fraction of at most 0.1% (see Berg’s celebrated book [92]) at which steric interactions are completely irrelevant. Furthermore, quorum-sensing and chemotaxis are the natural interactions encountered in most of the biophysics literature. The seminal Keller-Segel paper has over 3500 citations [Keller, Segell, Journal of theoretical biology 26, 399-415 (1970)], Miller & Bassler’s review on quorum-sensing reaches over 6000 citations [Miller, Bassler, Annual Reviews in Microbiology 55, 165-199 (2001)]. The communities interested in these interactions are very large!

That said, we agree with the referee that there are very interesting conditions in which pairwise forces are relevant, for instance for swarming bacterial colonies. To address the point of the referee, we have thus simulated a system in which particles both experience pairwise repulsive forces—the same between all species—and quorum-sensing interactions. This system is detailed in section 5 of the Supplementary Information. As shown in SI Movie 6, the four types of patterns reported in the article (MIPS, demixing, triple-phase coexistence and dynamical patterns) are observed. The tools to carry out the coarsening of these models have been recently derived, but they are even more involved than the ones used for QSAPs. While this is clearly on our todo-list, it goes beyond the scope of this work but is commented upon at the end of our article.

We hope that the discussion above and the new data on systems with pairwise forces will convince the referee that the scope of our results is broad enough to interest a large audience.

I have a few more comments:

1. *Eq. (6) results from a definition of the entropy production rate, which should certainly appear in the main text since sigma is the crucial quantity here.*

Response: We thank the referee for this comment and we now define σ in the main text.

Eq. (6) uses steady-state distributions. Does this mean all what the authors derive here is only valid in steady state?

Response: It is not clear to us what “all” refers to in this sentence and we thus discuss below the regimes in which all our results apply.

First, the fluctuating hydrodynamics that we construct predicts both the steady state of the system and the large-scale long-time relaxation dynamics. We have now clarified this in the article.

Regarding reversibility and the vanishing of the entropy production rate, we stress that the reversibility of *any* stochastic dynamics, even the equilibrium Langevin dynamics of a single particle in an external potential, is always assessed in the steady state. In equilibrium, this can be seen by considering the relaxation from any given distribution to the Boltzmann weight: such a relaxation is clearly not time-reversal symmetric. Here, the situation is the same: when $\sigma = 0$, the dynamics will first experience an irreversible transient until it converges to the steady state. Then, trajectories played forward or backward in time will be statistically undistinguishable. We now provide references both to a review on irreversibility in active matter [1], the review by Seifert on stochastic thermodynamics [64] and to some recent works of the Cates group that discuss these questions in detail [65-67].

2. *Certainly more information on Eq. (10) is needed. Why is it constructed like this.*

Response: We now detail the choice of Eq. (10) (which has become Eq. (11)). As we explained above this equation, this choice is motivated by biological systems in which regulations are implemented by different orthogonal QS circuits. However, we stress to the referee that this choice is not crucial—hence the initial lack of emphasis. We simply wanted a model to demonstrate our theoretical results, which are completely general.

3. Fig. 1 needs to be better discussed/explained.

Response: We thank the referee for raising this point and we now provide an extended discussion of the figure in its caption.

4. Again: Eq. (14) needs more explanation.

Response: We have detailed the construction of the former Eq (14) and now also refer to the explanations given for the former Eq (10). Again, this particular choice is convenient, but not crucial.

Report of reviewer #3

Summary

The authors study the collective behavior of self-propelled particles with non-reciprocal quorum-sensing interactions. They derive the fluctuating hydrodynamics of their microscopic model [eq.(3)], which allows them to provide a series of predictions. First, they obtain an explicit condition for an equilibrium mapping (EM) to exist at hydro level [eq.(7)], associated with vanishing entropy production rate. Within this mapping, they predict the phase diagram, and confirm its validity with particle-based simulations [figs.1-2]. Second, when the EM does not hold, they provide an explicit criterion for travelling patterns to appear [eq.(12)], and confirm again its validity with particle-based simulations [fig.3]. Finally, they consider non-reciprocity in chemotaxis, and show how the condition for EM translates in this context [eq.(17)]

Review

The manuscript is well written, and provides a very nice addition to the existing literature on non-reciprocal interactions for active mixtures. Importantly, the EM provides an explicit constraint for when microscopic non-reciprocity becomes irrelevant at hydrodynamic level. This constraint clearly demonstrates how coarse-graining methods can delineate when the minimization of an effective free energy is indeed legitimate to predict instabilities. Moreover, the agreement between predictions and simulations is indeed excellent, for all cases (even beyond the regime of effective free energy), without any fitting parameters.

Therefore, I recommend publication of the manuscript in Nature Communications. Please find below some minor comments that the authors might want to consider.

Response: We thank the referee for this precise and positive assessment of our work.

1. In the introduction, the authors justify non-reciprocity in active systems by arguing that they 'exchange momentum with their environment'. It is unclear why this condition should actually open the door to non-reciprocal interactions. Maybe, the authors could clarify their statement by arguing why, in contrast, passive systems do not typically 'exchange momentum with their environment' (if they do not).

Response: We thank the referee for this remark, which echoes a similar comment from Referee 1. This helped us realize that we were being too vague in the introduction, and that a clarification was needed. As explained in the reply to referee 1, the logic is as follows: activity requires exchanging momentum with a substrate/solvent. In turn, the latter will generically mediate interactions between particles. Because mediated interactions need not be reciprocal, we expect non-reciprocity to be generic in active systems.

The suggestion of the referee to discuss the passive case is interesting, but quite complex for people outside the field. We are afraid that this discussion would hinder rather than facilitate the reading of the article. Indeed, for passive particles in contact with a bath, when one integrates out the bath's degrees of freedom, the existence of a free energy functional for the particle system ensures effective reciprocal interactions. Proving this starting from microscopic models is actually a difficult task that was done, for instance, by Mazur and co-workers in the 70s (see, e.g., Bedeaux and Mazur, *Physica* 76 (1974) 247-258). This is a fascinating (and difficult) topic, which is too far from our article to be described at length and we have thus simply rephrased our formulation in terms of mediated interactions. Furthermore, when we introduce active particles interacting via quorum sensing, we now show explicitly that the resulting interactions are non-reciprocal. (See the new Eq. (2)). This should also help to make this discussion more precise.

2. In the introduction, it could be worth providing explicit references for each of the three types of self-propulsion considered here: active Brownian particles, run-and-tumble particles, and active Ornstein-

Uhlenbeck particles.

Response: We thank the referee for this suggestion, which we have followed.

Likewise, when mentioning the entropy production rate (and also around eq.(6)), it could be useful to provide more context and references: How is it defined? How is it connected to heat and irreversibility?

Response: We have now provided the definition of the entropy-production rate together with more references. We also stress that, at this coarse-grained level, the entropy-production rate is simply an information-based measurement that quantifies irreversibility and cannot generically be connected to heat. We have cited the recent works of the Cates group [65-67] that discuss this point in detail.

3. In the conclusion, the authors could mention other attempts to coarse-grain the dynamics, and to find an equilibrium mapping, for self-propelled particles beyond the case of quorum-sensing interactions. Indeed, for pair-wise interactions, analyzing the dynamics of the density field can prove more challenging. Here are recent attempts: J Stat Mech 2017, 113208; New J Phys 23, 103024 (2021); arXiv:2301.12155.

Response: We thank the referee for this suggestion. To address a comment by Referee 2, we now discuss the case of pairwise forces in the article and cite (among others) the references suggested by the referee.

1 Non-reciprocity across scales in active mixtures

2 A. Dinelli¹, J. O’Byrne^{1,2}, A. Curatolo³, Y. Zhao⁴, P. Sollich^{5,6}, J. Tailleur^{1,7,†}

¹*Université Paris Cité, Laboratoire Matière et Systèmes Complexes (MSC), UMR 7057 CNRS,*
*F-75205, 75205 Paris, France*
²*Department of Applied Maths and Theoretical Physics, University of Cambridge, Centre for*
*Mathematical Sciences, Wilberforce Rd, Cambridge CB3 0WA, UK*
³*John A. Paulson School of Engineering and Applied Sciences and Kavli Institute for Bionano*
*Science and Technology, Harvard University, Cambridge, MA 02138, USA*
⁴*Center for Soft Condensed Matter Physics and Interdisciplinary Research & School of Physical*
*Science and Technology, Soochow University, 215006 Suzhou, China*
⁵*Institute for Theoretical Physics, Georg-August-Universität Göttingen, 37 077 Göttingen, Ger-*
*many*
⁶*Department of Mathematics, King’s College London, London WC2R 2LS, UK*
⁷*Department of Physics, Massachusetts Institute of Technology, Cambridge, Massachusetts 02139,*
*USA*
† Corresponding authors: jgt@mit.edu
**In active matter, ~~the lack of momentum conservation makes non-reciprocal interactions the~~**
**~~rule rather than the exception. They lead~~ particles typically experience mediated interactions,**
**which are not constrained by Newton’s third law and are therefore generically non-reciprocal.**
Non-reciprocity leads to a rich set of emerging behaviors that are hard to account for **and**
~~to predict~~ starting from the microscopic scale, due to the absence of a generic theoretical
framework out of equilibrium. Here we consider bacterial mixtures that interact via me-
diated, non-reciprocal interactions (NRI) like quorum-sensing and chemotaxis. By explic-
ity relating microscopic and macroscopic dynamics, we show that, under conditions that we
derive explicitly, non-reciprocity may fade ~~as upon~~ coarse-graining ~~proceeds~~, leading to large-
scale ~~bona fide~~ equilibrium descriptions. In turns, this allows us to account quantitatively,
and without fitting parameters, for the rich behaviors observed in microscopic simulations
including phase separation, demixing ~~or~~, and multi-phase coexistence. We also derive the
condition under which non-reciprocity ~~is strong enough to survive~~ survives coarse-graining,
leading to a wealth of dynamical patterns. Again, ~~the explicit coarse-graining of the dynamics~~
our analytical approach allows us to predict the phase diagram of the system starting from
its microscopic description. All in all, our work demonstrates that the fate of non-reciprocity
across scales is a subtle and important question.
1 Introduction

[revised manuscript text omitted]
 = \nu$ and $\mu \neq \nu$, respectively. ~~We consider self-inhibition of motility coupled to a global
 165 enhancement of motility through~~

$$\underline{v_\mu(\mathbf{r}) = v_\mu^0 \phi_\mu^s[\tilde{\rho}_\mu(\mathbf{r})] \phi_\mu^g[\tilde{\rho}_t(\mathbf{r})] .}$$

166 ~~In Eq. , self and global regulations are modelled by sigmoidal functions ϕ_μ^s and ϕ_μ^g , respectively,
 167 and $\tilde{\rho}(\mathbf{r}) = K * \rho(\mathbf{r})$ is~~

168 We first consider self-regulation of motility by making v_μ depend on a local measurement
 169 of the density field obtained by convolution with a kernel of the same species, $\tilde{\rho}_\mu(\mathbf{r}) = K * \rho_\mu(\mathbf{r})$,
 170 where K (see Methods for details). Note that QS interactions leading to Eq. can actually be
 171 realized using is a bell-shaped function and the asterisk indicates a convolution of this kernel with
 172 the density field. We thus write $v_\mu(\mathbf{r}) = \hat{v}_\mu^0 \phi_\mu^s[\tilde{\rho}_\mu(\mathbf{r})]$, where ϕ_μ^s is an arbitrary function. In addition,
 173 we consider a second quorum-sensing circuit that makes the motility depend on the global density,
 174 so that \hat{v}_μ^0 is itself a functional of ρ_t . All in all, the self-propulsion speed of species μ thus reads

[revised manuscript text omitted]

217 **4 When non-reciprocal interactions survive ~~coarsening~~coarse-graining**

218 The violation of Eq. (3) is a sufficient condition for the emergence of non-reciprocal couplings
 219 between the density fields at the macroscopic scale (6). Consequently, the steady-state entropy
 220 production rate (7) is positive ~~-. The lack of and the hydrodynamic equations do not have~~ a gra-
 221 dient structure ~~for hydrodynamic equations- . In turn, it is well known to allow that this allows~~
 222 for the existence of travelling patterns ~~16,17,21-23,29,30,74 . To determine the , which can be studied~~

[revised manuscript text omitted]

658 *National Academy of Sciences* **118**, e2024083118 (2021).

Supplementary Material for: "Non-reciprocity across scales in active mixtures"

A. Dinelli,¹ J. O'Byrne,^{1,2} A. Curatolo,³ Y. Zhao,⁴ P. Sollich,^{5,6} and J. Tailleur^{†,1,7}

¹*Université Paris Cité, Laboratoire Matière et Systèmes Complexes (MSC), UMR 7057 CNRS, F-75205, 75205 Paris, France*

²*Department of Applied Maths and Theoretical Physics, University of Cambridge, Centre for Mathematical Sciences, Wilberforce Rd, Cambridge CB3 0WA, UK*

³*John A. Paulson School of Engineering and Applied Sciences and Kavli Institute for Bionano Science and Technology, Harvard University, Cambridge, MA 02138, USA*

⁴*Center for Soft Condensed Matter Physics and Interdisciplinary Research & School of Physical Science and Technology, Soochow University, 215006 Suzhou, China*

⁵*Department of Mathematics, King's College London, London WC2R 2LS, UK*

⁶*Institute for Theoretical Physics, Georg-August-Universität Göttingen, 37 077 Göttingen, Germany*

⁷*Department of Physics, Massachusetts Institute of Technology, Cambridge, Massachusetts 02139, USA*

(Dated: July 17, 2023)

† Corresponding author: jgt@mit.edu

CONTENTS

I. Coarse-graining the dynamics of active mixtures in the presence of mediated interactions	1
A. QS active mixtures: from microscopic dynamics to the mesoscopic Langevin equation	2
B. Derivation of the fluctuating hydrodynamics	4
C. Chemotactic mixtures from micro to macro	5
II. Generalized Schwarz theorem and functional integrability	7
III. Entropy production rate	8
IV. Coexisting densities in active mixtures	9
A. Three-phase coexistence	9
B. Two-phase coexistence	11
V. Mixture of QSAPs with pairwise forces	11
VI. Supplementary Movies	12
VII. Supplementary Figures	13
References	14

In Section I we detail the coarse-graining procedure to determine the fluctuating hydrodynamics describing the stochastic evolution at the macroscopic scale of mixtures of active particles in the presence of mediated interactions. We first derive the mesoscopic diffusion approximation of the RTP dynamics in Section IA with QS interactions and then derive the macroscopic field theory for the density fields ρ_μ in Section IB. We extend the derivation to chemotactic mixtures in Section IC. In Section II, we derive the generalized Schwarz theorem for N coupled stochastic field equations, which yields a practical condition under which our fluctuating hydrodynamics admits a generalized free energy. In Section III we compute the entropy production rate σ at the macroscopic level, and show that any microscopic violation of our generalized action-reaction principle leads to a positive entropy production rate at the macroscopic scale. Section IV is devoted to the construction of the phase diagram for the active mixtures that admit a generalized free energy at the macroscopic scale. We detail the derivation of the coexisting densities for the three-phase and two-phase coexistence regions. In Section VI we give the numerical details of the simulations of dynamical patterns reported in the supplementary movies. Finally, our supplementary figures can be found in Section VII.

I. COARSE-GRAINING THE DYNAMICS OF ACTIVE MIXTURES IN THE PRESENCE OF MEDIATED INTERACTIONS

In this section we present the coarse-graining procedure to obtain the fluctuating hydrodynamics (4) in the main text. For simplicity, we present here the full calculation for run-and-tumble particles, but the derivation below can be easily generalized to ABPs and AOUPs [1, 2]. ~~We refer the interested reader to a future paper for a complete review of coarse-graining methods in scalar active matter [?].~~

A. QS active mixtures: from microscopic dynamics to the mesoscopic Langevin equation

First, we consider the case of active mixtures of RTPs with QS interactions. Following the main text, $\mathbf{r}_{i,\mu}$ is the position of the i -th particle of species μ , with $\mu \in \{1, \dots, N\}$, and $\mathbf{u}_{i,\mu} \in \mathbb{S}^{d-1}$ is the unit vector describing the instantaneous orientation of the particle. For the sake of generality, we consider translational diffusion so that the dynamics read as

$$\dot{\mathbf{r}}_{i,\mu}(t) = v_\mu(\mathbf{r}_{i,\mu}, [\{\rho_\nu\}]) \mathbf{u}_{i,\mu} + \sqrt{2D_t} \eta_{i,\mu}(t) \quad (\text{I.1})$$

$$\mathbf{u} \rightarrow \mathbf{u}' \quad \text{where } \mathbf{u}' \text{ is drawn uniformly on the unit sphere } \mathbb{S}^{d-1} \text{ at rate } \tau_\mu^{-1}(\mathbf{r}, [\{\rho_\nu\}]) . \quad (\text{I.2})$$

The $\eta_{i,\mu}$ form a family of independent centered Gaussian white noises with unit variance.

Motility parameters are allowed to depend explicitly on the density fields, coupling the motion of all particles. However, since the density fields $\{\rho_\mu\}$ are conserved, their evolution occurs on large, diffusive timescales $T \sim L^2$, where L is the linear system size. When studying the particle dynamics on temporal scales $\tau_\mu \ll t \ll L^2$, we can therefore assume the density fields to be inhomogeneous but fixed, and only account for their dynamics on much longer time scales, for $t \sim L^2$. We thus first consider the case of non-interacting RTPs with position-dependent motility parameters:

$$\tau_\mu(\mathbf{r}_{i,\mu}, [\{\rho_\nu\}]), v_\mu(\mathbf{r}_{i,\mu}, [\{\rho_\nu\}]) \longrightarrow \tau_\mu(\mathbf{r}_{i,\mu}), v_\mu(\mathbf{r}_{i,\mu}) \quad (\text{I.3})$$

Let us now average out the fast, orientational degrees of freedom for the dynamics of a single particle i of species μ . Let $\mathcal{P}_\mu(\mathbf{r}, \mathbf{u}; t)$ be the probability of finding it at position \mathbf{r} with orientation \mathbf{u} at time t . The associated master equation reads:

$$\partial_t \mathcal{P}_\mu(\mathbf{r}, \mathbf{u}) = -\nabla_{\mathbf{r}} \cdot [v_\mu(\mathbf{r}) \mathbf{u} \mathcal{P}_\mu - D_t \nabla_{\mathbf{r}} \mathcal{P}_\mu] - \frac{1}{\tau_\mu} \mathcal{P}_\mu + \frac{1}{\Omega \tau_\mu} \int \mathcal{P}_\mu d\mathbf{u} \quad (\text{I.4})$$

where Ω is the area of \mathbb{S}^{d-1} . The dependence of \mathcal{P}_μ on \mathbf{u} can be expanded in harmonic tensors [3–5]:

$$\mathcal{P}_\mu(\mathbf{r}, \mathbf{u}) = \sum_{p=0}^{\infty} \frac{1}{\Omega} \frac{(d-2+2p)!!}{p!(d-2)!!} \mathbf{a}_\mu^p(\mathbf{r}) \cdot \widehat{\mathbf{u}}^{\otimes p} \quad (\text{I.5})$$

where $n!! \equiv \prod_{k=0}^{\lfloor (n-1)/2 \rfloor} (n-2k)$ is the double factorial, \mathbf{a}_μ^p the p^{th} -order harmonic component of \mathcal{P}_μ given by

$$\mathbf{a}_\mu^p(\mathbf{r}) = \int_{\mathbb{S}^{d-1}} \mathcal{P}_\mu(\mathbf{r}, \mathbf{u}) \widehat{\mathbf{u}}^{\otimes p} d\mathbf{u} , \quad (\text{I.6})$$

and $\widehat{\mathbf{u}}^{\otimes p}$ the traceless, symmetric part of the tensor $\mathbf{u}^{\otimes p}$. To carry out our coarse-graining procedure we will only need the explicit expressions of $\widehat{\mathbf{u}}^{\otimes p}$ for $p = 1, 2$ and 3, which are given by

$$\widehat{\mathbf{u}} = \mathbf{u} , \quad \widehat{\mathbf{u}}^{\otimes 2} = \mathbf{u}^{\otimes 2} - \frac{\mathbf{I}}{d} , \quad \text{and} \quad \widehat{\mathbf{u}}^{\otimes 3} = \mathbf{u}^{\otimes 3} - \frac{3}{d+2} \mathbf{u} \odot \mathbf{I} , \quad (\text{I.7})$$

where \mathbf{I} is the identity two-by-two tensor and $\mathbf{u} \odot \mathbf{I}$ the symmetrized version of $\mathbf{u} \otimes \mathbf{I}$, whose components in any orthonormal basis are $[\mathbf{u} \odot \mathbf{I}]^{\alpha\beta\gamma} = (u^\alpha \delta^{\beta\gamma} + u^\beta \delta^{\alpha\gamma} + u^\gamma \delta^{\alpha\beta})/3$.

To average over the (fast) orientational degrees of freedom, we integrate Eq. (I.4) with respect to \mathbf{u} to obtain the dynamics of $\mathbf{a}_\mu^0(\mathbf{r})$, the marginal with respect to \mathbf{u} of $\mathcal{P}_\mu(\mathbf{r}, \mathbf{u})$:

$$\partial_t \left\{ \int \mathcal{P}_\mu(\mathbf{r}, \mathbf{u}) d\mathbf{u} \right\} = -\nabla_{\mathbf{r}} \cdot [v_\mu \left\{ \int \mathbf{u} \mathcal{P}_\mu(\mathbf{r}, \mathbf{u}) d\mathbf{u} \right\} - D_t \nabla_{\mathbf{r}} \left\{ \int \mathcal{P}_\mu(\mathbf{r}, \mathbf{u}) d\mathbf{u} \right\}] . \quad (\text{I.8})$$

Using the definition (I.6) of $\widehat{\mathbf{a}}_\mu^0$ and \mathbf{a}_μ^1 , Eq. (I.8) becomes

$$\partial_t \mathbf{a}_\mu^0 = -\nabla_{\mathbf{r}} \cdot [v_\mu \mathbf{a}_\mu^1 - D_t \nabla_{\mathbf{r}} \mathbf{a}_\mu^0]. \quad (\text{I.9})$$

Equation (I.9) is not closed since it involves \mathbf{a}_μ^1 , the harmonic of order 1. To obtain the dynamics of the latter, we multiply Eq. (I.4) by \mathbf{u} and integrate over \mathbf{u} , yielding

$$\partial_t \mathbf{a}_\mu^1 = -\nabla_{\mathbf{r}} \cdot [v_\mu (\int \mathbf{u}^{\otimes 2} \mathcal{P}_\mu d\mathbf{u}) - D_t \nabla_{\mathbf{r}} \mathbf{a}_\mu^1] - \tau_\mu^{-1} \mathbf{a}_\mu^1. \quad (\text{I.10})$$

Since $\mathbf{u}^{\otimes 2} = \widehat{\mathbf{u}}^{\otimes 2} + \mathbf{I}/d$, the dynamics of \mathbf{a}_μ^1 can also be written as:

$$\partial_t \mathbf{a}_\mu^1 = -\nabla_{\mathbf{r}} \cdot [v_\mu (\mathbf{a}_\mu^2 + \frac{1}{d} \mathbf{I} \mathbf{a}_\mu^0) - D_t \nabla_{\mathbf{r}} \mathbf{a}_\mu^1] - \tau_\mu^{-1} \mathbf{a}_\mu^1. \quad (\text{I.11})$$

Similarly, one gets the dynamics of the second order harmonic moment \mathbf{a}_μ^2 —which appears in the dynamics (I.11) of \mathbf{a}_μ^1 —by multiplying Eq. (I.4) by $\widehat{\mathbf{u}}^{\otimes 2}$ and integrating with respect to \mathbf{u} :

$$\partial_t \mathbf{a}_\mu^2 = -\nabla_{\mathbf{r}} \cdot [v_\mu (\int \mathbf{u} \otimes \widehat{\mathbf{u}}^{\otimes 2} \mathcal{P}_\mu d\mathbf{u}) - D_t \nabla_{\mathbf{r}} \cdot \mathbf{a}_\mu^2] - \tau_\mu^{-1} \mathbf{a}_\mu^2. \quad (\text{I.12})$$

The explicit expressions of $\widehat{\mathbf{u}}^{\otimes 2}$ and $\widehat{\mathbf{u}}^{\otimes 3}$ given in Eq. (I.7) lead to

$$\begin{aligned} \mathbf{u} \otimes \widehat{\mathbf{u}}^{\otimes 2} &= \mathbf{u}^{\otimes 3} - \frac{1}{d} \mathbf{u} \otimes \mathbf{I} \\ &= \widehat{\mathbf{u}}^{\otimes 3} + \frac{3}{d+2} \mathbf{u} \odot \mathbf{I} - \frac{1}{d} \mathbf{u} \otimes \mathbf{I}. \end{aligned}$$

Thus, Eq. (I.12) can be re-written as:

$$\partial_t \mathbf{a}_\mu^2 = -\nabla_{\mathbf{r}} \cdot [v_\mu (\mathbf{a}_\mu^3 + \frac{3}{d+2} \mathbf{a}_\mu^1 \odot \mathbf{I} - \frac{1}{d} \mathbf{a}_\mu^1 \otimes \mathbf{I}) - D_t \nabla_{\mathbf{r}} \cdot \mathbf{a}_\mu^2] - \tau_\mu^{-1} \mathbf{a}_\mu^2. \quad (\text{I.13})$$

To close this hierarchy, we note that, on the one hand, \mathbf{a}_μ^0 is a conserved field (see Eq. (I.9)) and its relaxation time thus diverges with the system size. On the other hand, higher-order harmonics undergo both large-scale transport dynamics ($\sim \nabla_{\mathbf{r}}$) and fast exponential relaxations (with finite relaxation times $\sim \tau_\mu$). On time scales much larger than τ_μ , and for large system sizes, these dynamics thus decouple and we can assume that, for $p \geq 1$, \mathbf{a}_μ^p relaxes quasistatically to values enslaved to $\mathbf{a}_\mu^0(\mathbf{r}, t)$. We thus set $\partial_t \mathbf{a}_\mu^{1,2} = 0$ in Eqs. (I.11) and (I.13) to get:

$$\mathbf{a}_\mu^2 = \mathcal{O}(\nabla_{\mathbf{r}}^2) \quad (\text{I.14})$$

$$\mathbf{a}_\mu^1 = -\frac{\tau_\mu}{d} \nabla_{\mathbf{r}} (v_\mu \mathbf{a}_\mu^0) + \mathcal{O}(\nabla_{\mathbf{r}}^2) \quad (\text{I.15})$$

Finally, we insert Eq. (I.15) into the dynamics of the zeroth harmonic, Eq. (I.9), and we truncate the latter after terms of order $\mathcal{O}(\nabla_{\mathbf{r}}^2)$. This provides a diffusion-drift approximation to the run-and-tumble dynamics and relies on the fact that large-scale hydrodynamic modes are assumed to satisfy $\nabla_{\mathbf{r}}^k \sim \frac{1}{L^k}$.

All in all, we obtain a Fokker-Planck equation for the marginalized probability $\mathbf{a}_\mu^0(\mathbf{r}_i, t)$:

$$\partial_t \mathbf{a}_\mu^0 = -\nabla_{\mathbf{r}_i} \cdot [\mathbf{V}_\mu \mathbf{a}_\mu^0 - D_\mu \nabla_{\mathbf{r}_i} \mathbf{a}_\mu^0] \quad (\text{I.16})$$

where we introduced the mesoscopic drift \mathbf{V} and diffusivity D :

$$\mathbf{V}_\mu = -\frac{v_\mu \nabla v_\mu}{d} \tau_\mu \quad D_\mu = \frac{v_\mu^2 \tau_\mu}{d} + D_t \quad (\text{I.17})$$

Now that we have integrated out the orientational degrees of freedom, we can study the large-scale interacting dynamics by restoring the dependence on the density fields. In particular, we can associate to the Fokker-Planck equation (I.16) a corresponding mesoscopic Itô-Langevin equation for a particle of species μ :

$$\dot{\mathbf{r}}_{i,\mu} = \mathbf{V}_\mu(\mathbf{r}_{i,\mu}, [\{\rho_\nu\}]) + \nabla_{\mathbf{r}_{i,\mu}} D_\mu(\mathbf{r}_{i,\mu}, [\{\rho_\nu\}]) + \sqrt{2D_\mu(\mathbf{r}_{i,\mu}, [\{\rho_\nu\}])} \xi_{i,\mu}(t), \quad (\text{I.18})$$

where the $\xi_{i,\mu}$ are centered Gaussian white noises with unit variance.

A final remark: having restored the dependence of v_μ, τ_μ on the density fields, the gradient of the diffusivity $\nabla_{\mathbf{r}_{i,\mu}} D_\mu(\mathbf{r}_{i,\mu}, [\{\rho_\nu\}])$ in principle acts both on the first variable and on the fields $[\{\rho_\nu\}]$ (since ρ_μ is affected by a change in $\mathbf{r}_{i,\mu}$):

$$\nabla_{\mathbf{r}_{i,\mu}} D_\mu(\mathbf{r}_{i,\mu}, [\{\rho_\nu\}]) = \nabla_1 D_\mu(\mathbf{r}_{i,\mu}, [\{\rho_\nu\}]) + \left[\nabla_{\mathbf{r}_{i,\mu}} \frac{\delta D_\mu(\mathbf{r}')}{\delta \rho_\mu(\mathbf{r}_{i,\mu})} \right]_{\mathbf{r}'=\mathbf{r}_{i,\mu}} \quad (\text{I.19})$$

where ∇_1 is the derivative with respect to the first variable. However, it can be shown [6] that the second term of Eq. (I.19), i.e. the Itô drift, vanishes in many cases of interest. In particular, for the cases considered in this Letter, in which the τ_μ are constants, the Itô drift vanishes since v_μ (and thus D_μ) is a function of effective densities $\tilde{\rho}_\mu$ obtained by convolving the particle density $\rho_\nu(\mathbf{r})$ with a kernel $K(\mathbf{r})$ that is symmetric around the origin:

$$v_\mu(\mathbf{r}_{i,\mu}, [\{\rho_\nu\}]) = v_\mu(\{\tilde{\rho}_\nu(\mathbf{r}_{i,\mu})\}), \quad \tilde{\rho}_\nu(\mathbf{r}) = \int d^d \mathbf{r}' K(\mathbf{r} - \mathbf{r}') \rho_\nu(\mathbf{r}') \quad (\text{I.20})$$

Direct algebra then shows that the second term in Eq. (I.19) is proportional to $\nabla K(0) = 0$. In the following, this contribution to the Itô drift thus always vanishes.

B. Derivation of the fluctuating hydrodynamics

Starting from the Langevin equation (I.18), we now construct the time-evolution of the density field of species μ :

$$\rho_\mu(\mathbf{r}, t) = \sum_i \delta(\mathbf{r} - \mathbf{r}_{i,\mu}(t)) \quad (\text{I.21})$$

where the sum is taken over all particles of species μ . This is a straightforward generalization of the single-species case [6], which we detail here for the sake of completeness. Applying the Itô formula to Eq. (I.21), one gets

$$\frac{d}{dt} \rho_\mu(\mathbf{r}, t) = \sum_{i=1}^{N_\mu} \nabla_{\mathbf{r}_{i,\mu}} \delta(\mathbf{r} - \mathbf{r}_{i,\mu}(t)) \cdot \dot{\mathbf{r}}_{i,\mu} + D_\mu(\mathbf{r}_{i,\mu}, [\{\rho_\nu\}]) \nabla_{\mathbf{r}_{i,\mu}}^2 \delta(\mathbf{r} - \mathbf{r}_{i,\mu}(t)) \quad (\text{I.22})$$

To simplify the notation, in the following we omit the $[\{\rho_\nu\}]$ dependence in D_μ, \mathbf{V}_μ , which is implicitly assumed throughout the derivation. The first term in Eq. (I.22) can be re-expressed as:

$$\sum_{i=1}^{N_\mu} \nabla_{\mathbf{r}_{i,\mu}} \delta(\mathbf{r} - \mathbf{r}_{i,\mu}(t)) \cdot \dot{\mathbf{r}}_{i,\mu} = \sum_{i=1}^{N_\mu} \nabla_{\mathbf{r}_{i,\mu}} \delta(\mathbf{r} - \mathbf{r}_{i,\mu}) \cdot \left(\mathbf{V}_\mu(\mathbf{r}_{i,\mu}) + \nabla_{\mathbf{r}_{i,\mu}} D_\mu(\mathbf{r}_{i,\mu}) + \sqrt{2D_\mu(\mathbf{r}_{i,\mu})} \xi_{i,\mu} \right) \quad (\text{I.23})$$

$$= - \sum_{i=1}^{N_\mu} \nabla_{\mathbf{r}} \delta(\mathbf{r} - \mathbf{r}_{i,\mu}) \cdot \left(\mathbf{V}_\mu(\mathbf{r}_{i,\mu}) + \nabla_{\mathbf{r}_{i,\mu}} D_\mu(\mathbf{r}_{i,\mu}) + \sqrt{2D_\mu(\mathbf{r}_{i,\mu})} \xi_{i,\mu} \right) \quad (\text{I.24})$$

$$= - \sum_{i=1}^{N_\mu} \nabla_{\mathbf{r}} \cdot \left[\delta(\mathbf{r} - \mathbf{r}_{i,\mu}) \left(\mathbf{V}_\mu(\mathbf{r}_{i,\mu}) + \nabla_{\mathbf{r}_{i,\mu}} D_\mu(\mathbf{r}_{i,\mu}) + \sqrt{2D_\mu(\mathbf{r}_{i,\mu})} \xi_{i,\mu} \right) \right] \quad (\text{I.25})$$

$$= - \sum_{i=1}^{N_\mu} \nabla_{\mathbf{r}} \cdot \left[\delta(\mathbf{r} - \mathbf{r}_{i,\mu}) \left(\mathbf{V}_\mu(\mathbf{r}) + \nabla_{\mathbf{r}} D_\mu(\mathbf{r}) + \sqrt{2D_\mu(\mathbf{r})} \xi_{i,\mu} \right) \right] \quad (\text{I.26})$$

$$= - \nabla_{\mathbf{r}} \cdot \left[\rho_\mu(\mathbf{r}, t) \left(\mathbf{V}_\mu(\mathbf{r}) + \nabla_{\mathbf{r}} D_\mu(\mathbf{r}) \right) + \sqrt{2D_\mu \rho_\mu(\mathbf{r}, t)} \boldsymbol{\Lambda}_\mu(\mathbf{r}, t) \right]. \quad (\text{I.27})$$

To go from Eq. (I.26) to (I.27), we have introduced a centered Gaussian white noise field with unit variance, $\boldsymbol{\Lambda}_\mu(\mathbf{r}, t)$, and noticed that the two first cumulants of $-\nabla_{\mathbf{r}} \cdot \sqrt{2D_\mu \rho_\mu(\mathbf{r}, t)} \boldsymbol{\Lambda}_\mu(\mathbf{r}, t)$ and $-\sum_{i=1}^{N_\mu} \nabla_{\mathbf{r}} \cdot \sqrt{2D_\mu(\mathbf{r}_{i,\mu})} \xi_{i,\mu}$ coincide. These two Gaussian processes are thus identical [7].

Analogously for the second term in Eq. (I.22):

$$\sum_{i=1}^{N_\mu} D_\mu(\mathbf{r}_{i,\mu}) \nabla_{\mathbf{r}_{i,\mu}}^2 \delta(\mathbf{r} - \mathbf{r}_{i,\mu}(t)) = \sum_{i=1}^{N_\mu} D_\mu(\mathbf{r}_{i,\mu}) \nabla_{\mathbf{r}}^2 \delta(\mathbf{r} - \mathbf{r}_{i,\mu}(t)) = \sum_{i=1}^{N_\mu} \nabla_{\mathbf{r}}^2 [\delta(\mathbf{r} - \mathbf{r}_{i,\mu}(t)) D_\mu(\mathbf{r}_{i,\mu})] \quad (\text{I.28})$$

$$= \sum_{i=1}^{N_\mu} \nabla_{\mathbf{r}}^2 [\delta(\mathbf{r} - \mathbf{r}_{i,\mu}(t)) D_\mu(\mathbf{r})] = \nabla_{\mathbf{r}}^2 [\rho_\mu(\mathbf{r}, t) D_\mu(\mathbf{r})] \quad (\text{I.29})$$

Finally, we insert the expressions (I.27), (I.29) into Eq. (I.22) to get the fluctuating hydrodynamics of the density fields:

$$\partial_t \rho_\mu = -\nabla_{\mathbf{r}} \cdot \left\{ \mathbf{V}_\mu(\mathbf{r}, [\{\rho_\nu\}]) \rho_\mu - D_\mu(\mathbf{r}, [\{\rho_\nu\}]) \nabla_{\mathbf{r}} \rho_\mu + \sqrt{2D_\mu(\mathbf{r}, [\{\rho_\nu\}])} \rho_\mu \Lambda_\mu(\mathbf{r}, t) \right\}, \quad (\text{I.30})$$

which is Eq. (3) of the main text. In the case where $D_t = 0$, the macroscopic drift can be re-written as $V_\mu = -D_\mu \nabla_{\mathbf{r}} \log v_\mu$. Finally, introducing $M_\mu \equiv D_\mu \rho_\mu$ allows us to express the macroscopic theory as N coupled generalized models B:

$$\partial_t \rho_\mu = \nabla_{\mathbf{r}} \cdot \left\{ M_\mu \nabla_{\mathbf{r}} \mathbf{u}_\mu + \sqrt{2M_\mu} \Lambda_\mu(\mathbf{r}, t) \right\}, \quad \mathbf{u}_\mu(\mathbf{r}, [\{\rho_\nu\}]) = \log v_\mu(\mathbf{r}, [\{\rho_\nu\}]) + \log \rho_\mu(\mathbf{r}) \quad (\text{I.31})$$

C. Chemotactic mixtures from micro to macro

In this section, we derive the fluctuating hydrodynamics for chemotactic mixtures of RTPs in d dimensions starting from the microscopic dynamics (V.2), to show how the method laid out in sections IA-IB generalizes beyond the case of QS. We consider a mixture of N species of RTPs, whose self-propulsion speed and tumbling rate are biased by n distinct chemical fields $c_p(\mathbf{r})$ through:

$$v_\mu = v_{0\mu} - \mathbf{u}_{i,\mu} \cdot \sum_{p=1}^n v_{1\mu}^p \nabla_{\mathbf{r}_{i,\mu}} c_p(\mathbf{r}), \quad \tau_\mu^{-1} = \tau_{0\mu}^{-1} + \mathbf{u}_{i,\mu} \cdot \sum_{p=1}^n (\tau_{1\mu}^p)^{-1} \nabla_{\mathbf{r}_{i,\mu}} c_p(\mathbf{r}), \quad (\text{I.32})$$

where all parameters $v_{*\mu}^p, \tau_{*\mu}^p$ are constant.

In the main text, we focus on the case in which the fields are produced by the particles before they diffuse and degrade. For completeness, we show here how this leads to equations like Eq. (18) of the main text. The dynamics of each chemical field $c_p(\mathbf{r})$ then reads:

$$\partial_t c_p(\mathbf{r}, t) = \sum_{\mu=1}^N \chi_\mu^p \rho_\mu(\mathbf{r}, t) - \Gamma_p c_p(\mathbf{r}, t) + D_p \nabla_{\mathbf{r}}^2 c_p(\mathbf{r}, t), \quad (\text{I.33})$$

where χ_μ^p is the production rate of the chemical p by particles from species μ , Γ_p is its degradation rate, and D_p is its diffusivity. Since the evolutions of the conserved density fields $\rho_\mu = \sum_{i=1}^{N_\mu} \delta(\mathbf{r} - \mathbf{r}_{i,\mu}(t))$ occur on diffusive time scales, $t \propto L^2$, the finite relaxation times Γ_p^{-1} make the chemical fields adiabatically adapt to the values of the density fields on these time scales. Setting $\partial_t c_p = 0$ then leads to screened Poisson equations:

$$\mathcal{L}_p c_p(\mathbf{r}, t) \equiv \left(\frac{D_p}{\Gamma_p} \nabla_{\mathbf{r}}^2 - 1 \right) c_p(\mathbf{r}, t) = - \sum_{\mu=1}^N \frac{\chi_\mu^p}{\Gamma_p} \rho_\mu(\mathbf{r}, t). \quad (\text{I.34})$$

Eq. (I.34) is then solved by $c_p(\mathbf{r}, [\{\rho_\nu\}]) = \sum_{\nu=1}^N \frac{\chi_\nu^p}{\Gamma_p} G_p * \rho_\nu(\mathbf{r})$, where $G_p(\mathbf{r})$ is the Green function associated with \mathcal{L}_p .

On the microscopic time-scales $L^2 \gg t \gg \tau_\mu$ on which the dynamics of the RTPs becomes effectively diffusive, the density fields are essentially frozen. As before, we thus first consider the case in which a particle of species μ evolves in chemical fields $c_p(\mathbf{r})$ that are static and inhomogeneous. This allows us to write down a single-particle Master equation for the probability

$\mathcal{P}_\mu(\mathbf{r}, \mathbf{u})$:

$$\begin{aligned} \partial_t \mathcal{P}_\mu = & -\nabla_{\mathbf{r}} \cdot [v_{0\mu} - \mathbf{u} \cdot \sum_{p=1}^n v_{1\mu}^p \nabla_{\mathbf{r}} c_p] \mathbf{u} \mathcal{P}_\mu - D_t \nabla_{\mathbf{r}} \mathcal{P}_\mu] \\ & - \left(\frac{1}{\tau_{0\mu}} + \mathbf{u} \cdot \sum_{p=1}^n \frac{1}{\tau_{1\mu}^p} \nabla_{\mathbf{r}} c_p \right) \mathcal{P}_\mu + \frac{1}{\Omega} \int \left(\frac{1}{\tau_{0\mu}} + \mathbf{u} \cdot \sum_{p=1}^n \frac{1}{\tau_{1\mu}^p} \nabla_{\mathbf{r}} c_p \right) \mathcal{P}_\mu d\mathbf{u} \end{aligned} \quad (\text{I.35})$$

where Ω is the area of the d -dimensional unit sphere \mathbb{S}^{d-1} as before.

The derivation then follows the structure of Sec. IA: we expand $\mathcal{P}_\mu(\mathbf{r}, \mathbf{u})$ in harmonic tensors, using Eq. (I.5), and write down the dynamics of the first few harmonics:

$$\partial_t \mathbf{a}_\mu^0 = -\nabla_{\mathbf{r}} \cdot [v_{0\mu} \mathbf{a}_\mu^1 - (\mathbf{a}_\mu^2 + \frac{\mathbf{I}}{d} \mathbf{a}_\mu^0) \cdot \sum_{p=1}^n v_{1\mu}^p \nabla_{\mathbf{r}} c_p - D_t \nabla_{\mathbf{r}} \mathbf{a}_\mu^0]. \quad (\text{I.36})$$

$$\begin{aligned} \partial_t \mathbf{a}_\mu^1 = & -\nabla_{\mathbf{r}} \cdot [v_{0\mu} (\mathbf{a}_\mu^2 + \frac{\mathbf{I}}{d} \mathbf{a}_\mu^0) - (\mathbf{a}_\mu^3 + \frac{3}{d+2} \mathbf{a}_\mu^1 \odot \mathbf{I}) \cdot \sum_{p=1}^n v_{1\mu}^p \nabla_{\mathbf{r}} c_p - D_t \nabla_{\mathbf{r}} \mathbf{a}_\mu^1] \\ & - \frac{\mathbf{a}_\mu^1}{\tau_{0\mu}} - (\mathbf{a}_\mu^2 + \frac{\mathbf{I}}{d} \mathbf{a}_\mu^0) \cdot \sum_{p=1}^n \frac{1}{\tau_{1\mu}^p} \nabla_{\mathbf{r}} c_p. \end{aligned} \quad (\text{I.37})$$

$$\begin{aligned} \partial_t \mathbf{a}_\mu^2 = & -\nabla_{\mathbf{r}} \cdot \left[v_{0\mu} (\mathbf{a}_\mu^3 + \frac{3}{d+2} \mathbf{a}_\mu^1 \odot \mathbf{I} - \frac{1}{d} \mathbf{a}_\mu^1 \otimes \mathbf{I}) - \left(\mathbf{a}_\mu^4 + \frac{6}{d+4} \mathbf{a}_\mu^2 \odot \mathbf{I} - \frac{1}{d} \mathbf{a}_\mu^2 \otimes \mathbf{I} \right. \right. \\ & \left. \left. + \frac{3\Omega}{d(d+2)} \mathbf{I}^{\odot 2} - \frac{\Omega}{d^2} \mathbf{I}^{\otimes 2} \right) \cdot \sum_{p=1}^n v_{1\mu}^p \nabla_{\mathbf{r}} c_p - D_t \nabla_{\mathbf{r}} \cdot \mathbf{a}_\mu^2 \right] - \frac{\mathbf{a}_\mu^2}{\tau_{0\mu}} \\ & - (\mathbf{a}_\mu^3 + \frac{3}{d+2} \mathbf{a}_\mu^1 \odot \mathbf{I} - \frac{1}{d} \mathbf{a}_\mu^1 \otimes \mathbf{I}) \cdot \sum_{p=1}^n \frac{1}{\tau_{1\mu}^p} \nabla_{\mathbf{r}} c_p. \end{aligned} \quad (\text{I.38})$$

Within the diffusion-drift approximation, retaining all terms up to $\mathcal{O}(\nabla_{\mathbf{r}}^2)$ leads to:

$$\mathbf{a}_\mu^k = \mathcal{O}(\nabla_{\mathbf{r}}^2), \quad k \geq 2 \quad (\text{I.39})$$

$$\mathbf{a}_\mu^1 = -\frac{v_{0\mu} \tau_{0\mu}}{d} \nabla_{\mathbf{r}} \mathbf{a}_\mu^0 - \frac{\tau_{0\mu}}{d} \mathbf{a}_\mu^0 \sum_{p=1}^n \frac{1}{\tau_{1\mu}^p} \nabla_{\mathbf{r}} c_p + \mathcal{O}(\nabla_{\mathbf{r}}^2) \quad (\text{I.40})$$

Inserting Eq. (I.40) into the dynamics (I.36) of the zeroth-order harmonics and neglecting all terms $o(\nabla_{\mathbf{r}}^2)$, we obtain the mesoscopic Fokker-Planck equation for $\mathbf{a}_\mu^0(\mathbf{r}_i, t)$:

$$\partial_t \mathbf{a}_\mu^0 = -\nabla_{\mathbf{r}_i} \cdot [\mathbf{V}_\mu \mathbf{a}_\mu^0 - D_\mu \nabla_{\mathbf{r}_i} \mathbf{a}_\mu^0] \quad (\text{I.41})$$

where the drift velocity \mathbf{V}_μ and diffusivity D_μ read:

$$\mathbf{V}_\mu = -\frac{\tau_{0\mu}}{d} \sum_{p=1}^n \left(\frac{v_{1\mu}^p}{\tau_{0\mu}} + \frac{v_{0\mu}}{\tau_{1\mu}^p} \right) \nabla_{\mathbf{r}_i} c_p, \quad \text{where} \quad D_\mu = \frac{v_{0\mu}^2 \tau_{0\mu}}{d} + D_t. \quad (\text{I.42})$$

As for QS, we can associate to the Fokker-Planck equation (I.41) a corresponding Itô-Langevin equation for a particle of species μ , whose form is again given by Eq. (I.18). Since the Langevin equation has the same form as in the QS case, we obtain again a fluctuating hydrodynamics of the form:

$$\partial_t \rho_\mu = -\nabla_{\mathbf{r}} \cdot \left\{ \mathbf{V}_\mu(\mathbf{r}, [\{\rho_\nu\}]) \rho_\mu - D_\mu(\mathbf{r}, [\{\rho_\nu\}]) \nabla_{\mathbf{r}} \rho_\mu + \sqrt{2D_\mu(\mathbf{r}, [\{\rho_\nu\}])} \Lambda_\mu(\mathbf{r}, t) \right\}, \quad (\text{I.43})$$

with the drift and diffusion coefficients given this time by Eq. (I.42). When $D_t = 0$, introducing again $M_\mu \equiv D_\mu \rho_\mu$ leads to:

$$\partial_t \rho_\mu = \nabla_{\mathbf{r}} \cdot \left\{ M_\mu \nabla_{\mathbf{r}} \rho_\mu + \sqrt{2M_\mu} \Lambda_\mu(\mathbf{r}, t) \right\}, \quad \mathbf{u}_\mu(\mathbf{r}, [\{\rho_\nu\}]) = \frac{1}{v_{0\mu}^2} \sum_{p=1}^n \left(\frac{v_{1\mu}^p}{\tau_{0\mu}} + \frac{v_{0\mu}}{\tau_{1\mu}^p} \right) c_p(\mathbf{r}, [\{\rho_\nu\}]) + \log \rho_\mu(\mathbf{r}). \quad (\text{I.44})$$

II. GENERALIZED SCHWARZ THEOREM AND FUNCTIONAL INTEGRABILITY

In this section, we derive the generalized Schwarz integrability criterion that amounts to the vanishing of the distributions $D_{\mu\nu}$ given in Eq. (7) of the main text. Derivatives and differentials have been generalized to infinite dimensional topological vector spaces in the mathematical literature (see, e.g., [8, 9] and references therein). For simplicity, we consider density fields that belong to the Banach space $\mathbb{F} = C_b^k(\Omega, \mathbb{R}^n)$, defined as the set of k -times continuously differentiable functions on an open subset $\Omega \subseteq \mathbb{R}^d$ whose derivatives up to order k are bounded in the supremum norm:

$$\|\boldsymbol{\rho}\|_{\mathbb{F}} \equiv \sum_{p=0}^k \sum_{i_1, \dots, i_p=1}^d \sup_{\mathbf{r} \in \Omega} \left| \frac{\partial^p \boldsymbol{\rho}(\mathbf{r})}{\partial r_{i_1} \dots \partial r_{i_p}} \right|. \quad (\text{II.1})$$

In Eq. (II.1), $|\cdot|$ stands for the Euclidean norm in \mathbb{R}^n . In practice, we only consider density profiles that live in the open, simply connected subset \mathbb{F}_0 such that $\forall \mathbf{r}, \mu, \rho_{\mu}(\mathbf{r}) > 0$.

Functionals $\mathcal{F}[\boldsymbol{\rho}]$ are then mappings between two Banach spaces. A practical generalization of the differential is then given by the Fréchet derivative [10, 11]: a functional $\mathcal{F} : \mathbb{F} \rightarrow \mathbb{R}$ is said to be (Fréchet) differentiable at $\boldsymbol{\rho} \in \mathbb{F}$ if there exists a continuous linear map $\delta\mathcal{F}[\boldsymbol{\rho}] : \mathbb{F} \rightarrow \mathbb{R}$ such that

$$\mathcal{F}[\boldsymbol{\rho} + \boldsymbol{\phi}] = \mathcal{F}[\boldsymbol{\rho}] + \delta\mathcal{F}[\boldsymbol{\rho}](\boldsymbol{\phi}) + o(\|\boldsymbol{\phi}\|_{\mathbb{F}}) \quad (\text{II.2})$$

for all $\boldsymbol{\phi} \in \mathbb{F}$. The differential at $\boldsymbol{\rho}$, $\delta\mathcal{F}[\boldsymbol{\rho}]$, applied to $\boldsymbol{\phi}$ is usually written as

$$\delta\mathcal{F}[\boldsymbol{\rho}](\boldsymbol{\phi}) = \sum_{\mu=1}^n \int_{\Omega} \frac{\delta\mathcal{F}[\boldsymbol{\rho}]}{\delta\rho_{\mu}(\mathbf{r})} \phi_{\mu}(\mathbf{r}) \, d\mathbf{r}, \quad (\text{II.3})$$

where $(\frac{\delta\mathcal{F}[\boldsymbol{\rho}]}{\delta\rho_1(\mathbf{r})}, \dots, \frac{\delta\mathcal{F}[\boldsymbol{\rho}]}{\delta\rho_n(\mathbf{r})})$ is the functional derivative of \mathcal{F} at $\boldsymbol{\rho}$.

Higher order derivatives are then defined in a recursive manner. For instance, a functional $\mathcal{F} : \mathbb{F} \rightarrow \mathbb{R}$ is twice differentiable if $\delta\mathcal{F} : \boldsymbol{\rho} \in \mathbb{F} \rightarrow \delta\mathcal{F}[\boldsymbol{\rho}](\cdot) \in L(\mathbb{F}, \mathbb{R})$ is Fréchet differentiable – $L(\mathbb{F}, \mathbb{R})$ being the Banach space of continuous linear maps from \mathbb{F} to \mathbb{R} . This Fréchet derivative, denoted by $\delta^2\mathcal{F}$, goes from \mathbb{F} to $L(\mathbb{F}, L(\mathbb{F}, \mathbb{R}))$. Since the latter space is isomorphic to $L(\mathbb{F} \times \mathbb{F}, \mathbb{R})$ for Banach spaces, one usually thinks about $\delta^2\mathcal{F}[\boldsymbol{\rho}](\cdot, \cdot)$ as a bilinear map. In this context, one can Taylor expand to second order a twice-differentiable functional:

$$\mathcal{F}[\boldsymbol{\rho} + \boldsymbol{\phi}] = \mathcal{F}[\boldsymbol{\rho}] + \delta\mathcal{F}[\boldsymbol{\rho}](\boldsymbol{\phi}) + \frac{1}{2} \delta^2\mathcal{F}[\boldsymbol{\rho}](\boldsymbol{\phi}, \boldsymbol{\phi}) + o(\|\boldsymbol{\phi}\|_{\mathbb{F}}^2). \quad (\text{II.4})$$

In this case, the second order differential is necessarily symmetric [12], *i.e.* it satisfies

$$\forall \boldsymbol{\phi}, \boldsymbol{\psi} \in \mathbb{F}, \quad \delta^2\mathcal{F}[\boldsymbol{\rho}](\boldsymbol{\phi}, \boldsymbol{\psi}) = \delta^2\mathcal{F}[\boldsymbol{\rho}](\boldsymbol{\psi}, \boldsymbol{\phi}). \quad (\text{II.5})$$

In terms of functional derivatives, this last equality reads

$$\sum_{\mu, \nu=1}^n \int \frac{\delta^2\mathcal{F}[\boldsymbol{\rho}]}{\delta\rho_{\mu}(\mathbf{r})\delta\rho_{\nu}(\mathbf{r}')} \phi_{\mu}(\mathbf{r})\psi_{\nu}(\mathbf{r}') \, d\mathbf{r}d\mathbf{r}' = \sum_{\mu, \nu=1}^n \int \frac{\delta^2\mathcal{F}[\boldsymbol{\rho}]}{\delta\rho_{\mu}(\mathbf{r}')\delta\rho_{\nu}(\mathbf{r})} \psi_{\mu}(\mathbf{r}')\phi_{\nu}(\mathbf{r}) \, d\mathbf{r}d\mathbf{r}'. \quad (\text{II.6})$$

Exchanging the dummy indices $\mu \rightarrow \nu$ and the dummy variables $\mathbf{r} \rightarrow \mathbf{r}'$ on the right hand side of Eq. (II.6) leads to

$$\sum_{\mu, \nu=1}^n \int \left(\frac{\delta^2\mathcal{F}[\boldsymbol{\rho}]}{\delta\rho_{\mu}(\mathbf{r})\delta\rho_{\nu}(\mathbf{r}')} - \frac{\delta^2\mathcal{F}[\boldsymbol{\rho}]}{\delta\rho_{\nu}(\mathbf{r}')\delta\rho_{\mu}(\mathbf{r})} \right) \phi_{\mu}(\mathbf{r})\psi_{\nu}(\mathbf{r}') \, d\mathbf{r}d\mathbf{r}' = 0. \quad (\text{II.7})$$

Equation (II.7) can then be rewritten as an equation for distributions:

$$\frac{\delta^2\mathcal{F}[\boldsymbol{\rho}]}{\delta\rho_{\mu}(\mathbf{r})\delta\rho_{\nu}(\mathbf{r}')} = \frac{\delta^2\mathcal{F}[\boldsymbol{\rho}]}{\delta\rho_{\nu}(\mathbf{r}')\delta\rho_{\mu}(\mathbf{r})}. \quad (\text{II.8})$$

Note that, in the symmetry relation Eq. (II.8), both indices μ, ν and continuous variables \mathbf{r}, \mathbf{r}' have to be exchanged simultaneously. Let us now consider a map $\mathbf{u}(\mathbf{r}, [\boldsymbol{\rho}]) = (u_1(\mathbf{r}, [\boldsymbol{\rho}]), \dots, u_n(\mathbf{r}, [\boldsymbol{\rho}]))$ that is a function of \mathbf{r} and a functional of $\boldsymbol{\rho}$. By

application of condition (II.8), a necessary condition for \mathbf{u} to be the functional derivative of some functional \mathcal{F} , *i.e.*

$$\mathbf{u}_\mu(\mathbf{r}, [\boldsymbol{\rho}]) = \frac{\delta \mathcal{F}[\boldsymbol{\rho}]}{\delta \rho_\mu}, \quad (\text{II.9})$$

is then:

$$\frac{\delta \mathbf{u}_\nu(\mathbf{r}', [\boldsymbol{\rho}])}{\delta \rho_\mu(\mathbf{r})} = \frac{\delta \mathbf{u}_\mu(\mathbf{r}, [\boldsymbol{\rho}])}{\delta \rho_\nu(\mathbf{r}')}, \quad (\text{II.10})$$

which is the integrability condition given in the main text. The sufficient condition then stems from the fact that \mathbb{F}_0 is simply connected.

Equation (II.10) is the generalization of the functional Schwarz theorem to functionals of vector-valued fields. It is interesting to note that, in the single-field case, the functional Schwarz theorem reads (see, e.g., [13]):

$$\frac{\delta u(\mathbf{r}', [\rho])}{\delta \rho(\mathbf{r})} = \frac{\delta u(\mathbf{r}, [\rho])}{\delta \rho(\mathbf{r}')}. \quad (\text{II.11})$$

Equation (II.10) is thus *not* the independent application of (II.11) to each component u_μ .

Finally, we note that the topology generated by the norm (II.1) on $C_b^k(\Omega)$ (which makes the latter a Banach space) allows one to treat (the linear map associated with) functional derivatives of the form

$$\frac{\delta}{\delta \rho_\mu(\mathbf{r}')} \frac{\partial^k \rho_\mu(\mathbf{r})}{\partial r_{i_1} \dots \partial r_{i_k}} = \frac{\partial \delta(\mathbf{r} - \mathbf{r}')}{\partial r_{i_1} \dots \partial r_{i_k}}. \quad (\text{II.12})$$

as proper Fréchet derivatives, since they become continuous with respect to that topology.

III. ENTROPY PRODUCTION RATE

In this section, we derive the expression of the entropy production rate for N coupled model-B dynamics, and show how the violation of Schwarz's condition for the existence of a free energy leads to a positive entropy production.

Let $\mathcal{P}[\{\rho_\nu(\mathbf{r}, t)\}; t_f]$ be the probability to observe a trajectory of the system in a time window $(0, t_f)$, in the steady state. The time-reversed trajectory then has a probability: $\mathcal{P}^\dagger[\{\rho_\nu(\mathbf{r}, t)\}; t_f] \equiv \mathcal{P}[\{\rho_\nu(\mathbf{r}, t_f - t)\}; t_f]$. The entropy production rate is defined as:

$$\sigma = \lim_{t_f \rightarrow \infty} \frac{1}{t_f} \log \frac{\mathcal{P}[\{\rho_\nu(\mathbf{r}, t)\}; t_f]}{\mathcal{P}^\dagger[\{\rho_\nu(\mathbf{r}, t)\}; t_f]} \quad (\text{III.1})$$

For the field dynamics defined by eq. (7) of the main text, the probability of a trajectory is easily obtained using a path integral representation [14]. Using Stratonovich time-discretization¹:

$$\sigma = \lim_{t \rightarrow \infty} \frac{1}{t} \left[- \int_0^t dt' \int d^d \mathbf{r} \sum_{\mu=1}^N \frac{(\mathbf{J}_\mu + M_\mu \nabla \mathbf{u}_\mu)^2}{4M_\mu} + \int_0^t dt' \int d^d \mathbf{r} \sum_{\mu=1}^N \frac{(-\mathbf{J}_\mu + M_\mu \nabla \mathbf{u}_\mu)^2}{4M_\mu} \right] \quad (\text{III.2})$$

where \mathbf{J}_μ is the real-space current: $\mathbf{J}_\mu = -M_\mu \nabla \mathbf{u}_\mu + \sqrt{2M_\mu} \boldsymbol{\Lambda}_\mu$. The stochastic current \mathbf{J}_μ is the only term that changes sign under time-reversal, which explains the minus sign in the second integral of (III.2). Direct calculations then lead to:

$$\sigma = \lim_{t \rightarrow \infty} \frac{1}{t} \int_0^t dt' \int d^d \mathbf{r} \sum_{\mu=1}^N (\nabla \cdot \mathbf{J}_\mu) \mathbf{u}_\mu = \lim_{t \rightarrow \infty} -\frac{1}{t} \int_0^t dt' \int d^d \mathbf{r} \sum_{\mu=1}^N \mathbf{u}_\mu \circ \dot{\rho}_\mu = - \int d^d \mathbf{r} \sum_{\mu=1}^N \langle \mathbf{u}_\mu \circ \dot{\rho}_\mu \rangle \quad (\text{III.3})$$

where the last equality relies on ergodicity to replace the time average with an ensemble average. The symbol \circ denotes the Stratonovich product.

¹ Equation (7) of the main text was obtained using Itô time discretization. Nonetheless, the stochastic hydrodynamics in Itô and Stratonovich would differ only by a vanishing Itô drift $\nabla_{\mathbf{r}_{i,\mu}} \frac{\delta D_{i,\mu}}{\delta \rho_\mu} = 0$, as remarked at the end of Sec. IA.

Next, we show explicitly that σ is always positive whenever the chemical potential cannot be derived from a free energy functional \mathcal{F} . To do so, we use the results of [15]—generalized to field theory—to rewrite the ensemble average in (III.3) as:

$$\sigma = - \int \mathcal{D}[\{\rho_\nu\}] \int d^d \mathbf{r} \sum_{\mu=1}^N \mathbf{u}_\mu[\mathbf{r}, \{\rho_\nu\}] \mathcal{J}_s^\mu[\mathbf{r}, \{\rho_\nu\}] \quad (\text{III.4})$$

where $\mathcal{J}_s^\mu(\mathbf{r}, [\{\rho_\nu\}])$ is the steady-state probability current in the functional Fokker-Planck equation for $P_s[\{\rho_\nu\}]$:

$$\mathcal{J}_s^\mu(\mathbf{r}, [\{\rho_\nu\}]) = \nabla \cdot [M_\mu \nabla \mathbf{u}_\mu + M_\mu \nabla \frac{\delta \log P_s}{\delta \rho_\mu(\mathbf{r})}] P_s. \quad (\text{III.5})$$

We then note that

$$\int \mathcal{D}[\{\rho_\nu\}] \int d^d \mathbf{r} \mathcal{J}_s^\mu \frac{\delta \log P_s}{\delta \rho_\mu(\mathbf{r})} = \int d^d \mathbf{r} \langle \frac{\delta \log P_s}{\delta \rho_\mu(\mathbf{r})} \circ \dot{\rho}_\mu \rangle = \lim_{\tau \rightarrow \infty} \frac{P_s[\{\rho_\nu(\tau)\}] - P_s[\{\rho_\nu(0)\}]}{\tau} = 0. \quad (\text{III.6})$$

This allows one to rewrite σ as

$$\sigma = - \int \mathcal{D}[\{\rho_\nu\}] \int d^d \mathbf{r} \sum_{\mu=1}^N (\mathbf{u}_\mu[\mathbf{r}, \{\rho_\nu\}] + \frac{\delta \log P_s}{\delta \rho_\mu(\mathbf{r})}) \mathcal{J}_s^\mu[\mathbf{r}, \{\rho_\nu\}] \quad (\text{III.7})$$

$$= - \int \mathcal{D}[\{\rho_\nu\}] \int d^d \mathbf{r} \sum_{\mu=1}^N (\mathbf{u}_\mu[\mathbf{r}, \{\rho_\nu\}] + \frac{\delta \log P_s}{\delta \rho_\mu(\mathbf{r})}) \nabla \cdot \{M_\mu [\nabla \mathbf{u}_\mu + \nabla \frac{\delta \log P_s}{\delta \rho_\mu(\mathbf{r})}] P_s\} \quad (\text{III.8})$$

$$= \int \mathcal{D}[\{\rho_\nu\}] P_s[\{\rho_\nu\}] \int d^d \mathbf{r} \sum_{\mu=1}^N M_\mu [\nabla (\mathbf{u}_\mu[\mathbf{r}, \{\rho_\nu\}] + \frac{\delta \log P_s}{\delta \rho_\mu(\mathbf{r})})]^2 \quad (\text{III.9})$$

$$= \langle \int d^d \mathbf{r} \sum_{\mu=1}^N M_\mu [\nabla (\mathbf{u}_\mu[\mathbf{r}, \{\rho_\nu\}] + \frac{\delta \log P_s}{\delta \rho_\mu(\mathbf{r})})]^2 \rangle \geq 0 \quad (\text{III.10})$$

Finally, one can generalize the norm introduced by Otto [16, 17]:

$$\|\mathbf{f}(\mathbf{r})\|^2 \equiv \int d^d \mathbf{r} \sum_{\mu=1}^N M_\mu [\nabla f_\mu(\mathbf{r})]^2 \quad (\text{III.11})$$

to rewrite σ as

$$\sigma = \langle \|\mathbf{u} + \frac{\delta \log P_s}{\delta \boldsymbol{\rho}}\|^2 \rangle \quad (\text{III.12})$$

which directly relates the entropy production rate to the non-conservative nature of \mathbf{u}_μ .

IV. COEXISTING DENSITIES IN ACTIVE MIXTURES

To make the results of our work self-contained, we briefly recall how phase coexistence in binary mixtures can be constructed from the knowledge of the free energy density. We refer the interested reader to [18] for more details. According to the Gibbs phase rule, in a binary mixture both three-phase and two-phase coexistence regions are possible.

A. Three-phase coexistence

Each phase i is characterized by the densities of the two components, which we denote using vector notation as $\boldsymbol{\rho}^i = (\rho_\alpha^i, \rho_\beta^i)$, with $i = 1, 2, 3$. These three points determine a triangle in the $(\rho_\alpha^0, \rho_\beta^0)$ -plane, which is the region of three-phase coexistence. Any initially homogeneous system with density $\boldsymbol{\rho}^0$ inside this region will separate into three phases whose compositions corre-

spond to the corners of the triangle. The fraction of the system corresponding to each phase is then set by the lever rule:

$$\boldsymbol{\rho}^0 = (1 - s - t)\boldsymbol{\rho}^1 + s\boldsymbol{\rho}^2 + t\boldsymbol{\rho}^3 \quad (\text{IV.1})$$

where $s \in [0, 1]$, $t \in [0, 1 - s]$ are the fractional volumes of phases 2 and 3, respectively. Pictorially, $\boldsymbol{\rho}^0$ can be seen as the position of the center of mass of a triangle whose vertices are located at $\boldsymbol{\rho}^1, \boldsymbol{\rho}^2, \boldsymbol{\rho}^3$ and have masses $(1 - s - t), s, t$ respectively (see Fig. S1). To determine the phase diagram, one thus only needs to determine the values of the $\boldsymbol{\rho}^i$.

In the large system-size limit, we can neglect the contributions of interfaces to the free energy so that, in the three-phase coexistence region, the free energy can be approximated as:

$$\mathcal{F} \simeq V[(1 - s - t)f(\boldsymbol{\rho}^1) + sf(\boldsymbol{\rho}^2) + tf(\boldsymbol{\rho}^3)] \quad (\text{IV.2})$$

We introduce two Lagrange multipliers u_α^0, u_β^0 , to minimize the free energy subject to the constraint of conservation of the total numbers of α and β particles. At this stage u_α^0 and u_β^0 are unknown constants, with the notation anticipating the identification below as chemical potentials. We thus need to minimize the function

$$\Psi \equiv V\{(1 - s - t)f(\boldsymbol{\rho}^1) + sf(\boldsymbol{\rho}^2) + tf(\boldsymbol{\rho}^3) - u_\alpha^0[(1 - s - t)\rho_\alpha^1 + s\rho_\alpha^2 + t\rho_\alpha^3] - u_\beta^0[(1 - s - t)\rho_\beta^1 + s\rho_\beta^2 + t\rho_\beta^3]\} \quad (\text{IV.3})$$

Differentiating with respect to $\boldsymbol{\rho}_{\alpha,\beta}^i$ we get:

$$\left\{ \begin{array}{l} \frac{\partial \Psi}{\partial \boldsymbol{\rho}_{\alpha,\beta}^1} = 0 \Rightarrow V[(1 - s - t)\nabla_\rho f(\boldsymbol{\rho}^1) - (1 - s - t)\mathbf{u}^0] = 0 \\ \frac{\partial \Psi}{\partial \boldsymbol{\rho}_{\alpha,\beta}^2} = 0 \Rightarrow V[s\nabla_\rho f(\boldsymbol{\rho}^2) - s\mathbf{u}^0] = 0 \\ \frac{\partial \Psi}{\partial \boldsymbol{\rho}_{\alpha,\beta}^3} = 0 \Rightarrow V[t\nabla_\rho f(\boldsymbol{\rho}^3) - t\mathbf{u}^0] = 0, \end{array} \right. \quad (\text{IV.4})$$

where we have introduced $\nabla_\rho \equiv (\frac{\partial}{\partial \rho_\alpha}, \frac{\partial}{\partial \rho_\beta})$ and $\mathbf{u}^0 \equiv (u_\alpha^0, u_\beta^0)$. Equation (IV.4) is solved by

$$\boxed{\nabla_\rho f(\boldsymbol{\rho}^1) = \nabla_\rho f(\boldsymbol{\rho}^2) = \nabla_\rho f(\boldsymbol{\rho}^3) = \mathbf{u}^0} \quad (\text{IV.5})$$

We can thus identify the Lagrange multipliers $u_{\alpha,\beta}^0$ with the *chemical potentials* $u_{\alpha,\beta}$ of each species:

$$\mathbf{u}_\mu = \frac{\partial F}{\partial N_\mu} = \frac{\partial f}{\partial \rho_\mu} \quad (\text{IV.6})$$

where we have used that, in a homogeneous system, $F = Vf(\boldsymbol{\rho})$ and $\rho_\mu = N_\mu/V$. Hence, Eq. (IV.5) establishes the equality of each species' chemical potential in the three phases. Finally, minimizing Ψ with respect to s, t leads to:

$$\left\{ \begin{array}{l} \frac{\partial \Psi}{\partial s} = 0 \Rightarrow -f(\boldsymbol{\rho}^1) + f(\boldsymbol{\rho}^2) - \mathbf{u}^0 \cdot (\boldsymbol{\rho}^2 - \boldsymbol{\rho}^1) = 0 \\ \frac{\partial \Psi}{\partial t} = 0 \Rightarrow -f(\boldsymbol{\rho}^1) + f(\boldsymbol{\rho}^3) - \mathbf{u}^0 \cdot (\boldsymbol{\rho}^3 - \boldsymbol{\rho}^1) = 0 \end{array} \right. \quad (\text{IV.7})$$

corresponding to the equality of *total pressure*:

$$\boxed{P = -\frac{\partial F}{\partial V} = \mathbf{u}^0 \cdot \boldsymbol{\rho}^1 - f(\boldsymbol{\rho}^1) = \mathbf{u}^0 \cdot \boldsymbol{\rho}^2 - f(\boldsymbol{\rho}^2) = \mathbf{u}^0 \cdot \boldsymbol{\rho}^3 - f(\boldsymbol{\rho}^3)} \quad (\text{IV.8})$$

In conclusion, we have found a *common tangent plane* to the free energy surface f . This plane is tangent to f in 3 points, i.e. the three coexisting densities, which delimit a triangular 3-phase coexistence region. The intercept of this plane with the f axis corresponds to $-P$, while the slopes along the α -axis and β -axis represent respectively u_α and u_β . In practice, to determine the properties of the three coexisting phases we solve (with a numerical solver) Eq. (IV.5)-(IV.8) in the variables $(\boldsymbol{\rho}^1, \boldsymbol{\rho}^2, \boldsymbol{\rho}^3)$, which makes a system of 6 independent equations for 6 unknowns.

Let us finally note that, in equilibrium, P and $u_{\alpha,\beta}$ are the usual thermodynamic pressure and chemical potentials. For active systems, one has to be careful since these effective chemical potentials and pressure do not have the same properties as for

equilibrium systems. For instance, P will not measure the force exerted by the system on a confining container [19].

B. Two-phase coexistence

When we apply the common tangent construction to find the two-phase coexistence regions, fewer constraints are present. Hence, instead of obtaining 3 points of coexisting densities in the $(\rho_\alpha^0, \rho_\beta^0)$ -plane as in the previous case, we get binodal curves of coexisting densities. A homogeneous system inside this region with a given $(\rho_\alpha^0, \rho_\beta^0)$ will separate between the corresponding liquid and gas phases along a *tie line*.

Repeating the extremization procedure described for the three-phase case then leads to:

$$\begin{cases} \nabla_\rho f_1 = \nabla_\rho f_2 = \mathbf{u} & (\text{Equality of chemical potentials}) \\ f_2 - \mathbf{u} \cdot \boldsymbol{\rho}^2 = f_1 - \mathbf{u} \cdot \boldsymbol{\rho}^1 & (\text{Equality of total pressure}) \end{cases} \quad (\text{IV.9})$$

If the system is initially in $(\rho_\alpha^0, \rho_\beta^0)$, two further constraints must be added:

$$\begin{cases} (1-s)\rho_\alpha^1 + s\rho_\alpha^2 = \rho_\alpha^0 \\ (1-s)\rho_\beta^1 + s\rho_\beta^2 = \rho_\beta^0 \end{cases} \quad s \in [0, 1] \text{ fractional volume} \quad (\text{IV.10})$$

For a given initial composition $(\rho_\alpha^0, \rho_\beta^0)$, solving Eq. (IV.9)-(IV.10) provides the species densities in the coexisting phases and the fractional volume s of the first phase. Varying $\boldsymbol{\rho}_0$, we are thus able to construct the full binodal lines.

V. MIXTURE OF QSAPS WITH PAIRWISE FORCES

To see how the results presented in this article can be generalized in the presence of pairwise forces, we simulate a mixture of two species of RTPs with soft repulsive interactions coupled to cross QS interactions. The dynamics of particle i of type μ reads:

$$\dot{\mathbf{r}}_{i,\mu}(t) = v_\mu(\mathbf{r}_{i,\mu}, [\{\rho_\nu\}]) \mathbf{u}_{i,\mu} - \sum_{(j,\nu) \neq (i,\mu)} \nabla_{\mathbf{r}_i} V(|\mathbf{r}_{(i,\mu)} - \mathbf{r}_{(j,\nu)}|) + \sqrt{2D_t} \eta_{i,\mu}(t) \quad (\text{V.1})$$

$$\mathbf{u} \xrightarrow{\tau^{-1}} \mathbf{u}' \quad \text{where } \mathbf{u}' \text{ is drawn uniformly on the unit circle.} \quad (\text{V.2})$$

We model our particles as soft repulsive spheres, using a repulsive truncated harmonic potential:

$$V(r) = \begin{cases} E_0(1-r^2) & \text{for } r < r_0 \\ 0 & \text{otherwise.} \end{cases}, \quad (\text{V.3})$$

For the QS cross-regulation of the self-propulsion speed, we choose

$$v_\mu(\tilde{\rho}_\nu) = v_0 \exp\left\{ \kappa_\nu^c \left[\tanh\left(\frac{\tilde{\rho}_\nu - \bar{\rho}}{\delta\rho}\right) + \tanh\left(\frac{\bar{\rho}}{\delta\rho}\right) \right] \right\}, \quad (\text{V.4})$$

where the local density fields are measured as

$$\tilde{\rho}_\nu(\mathbf{r}) = \int_{|\mathbf{r}-\mathbf{r}'| < r_{QS}} d^d \mathbf{r}' K(\mathbf{r} - \mathbf{r}') \rho_\nu(\mathbf{r}'), \quad (\text{V.5})$$

as in the rest of our simulations.

In SI Movie 6, we show that the phenomenology of this model is akin to the one reported for mixtures of QS active particles without steric repulsion. In the absence of QS interactions, we observe the standard motility-induced phase separation (SI Movie 6-a). In the presence of cross-enhancement of motility, we report both demixed (SI Movie 6-b) and triple-coexistence phases (SI Movie 6-c), while opposite cross-interactions lead to dynamical patterns (SI Movie 6-d). The parameters of SI Movie 6 are

discussed in Section VI, together with those of SI Movies 1-5 that correspond to the simulations reported in Figs 1 & 3 of the main text.

VI. SUPPLEMENTARY MOVIES

In this Section, we first provide the parameters of the five supplementary movies that illustrate the variety of dynamical patterns observed numerically in mixtures of active particles interacting via quorum-sensing. The self-propulsion speeds v_μ used in the simulations are given by Eqs. (14) and (21) of the main text. The following parameters are common to all simulations: system size 30×30 , $dt = 0.005$, $\bar{\rho} = 25$, $\delta\rho = 10$, $v_{\alpha,\beta}^0 = 5$, $\tau_{\alpha,\beta} = 1$, $\kappa_{\alpha,\beta}^s = -1$, $\rho_{\alpha,\beta}^0 = 25$. The amplitudes of the cross interactions, $\kappa_{\alpha,\beta}^c$, have opposite signs and their specific values change from one movie to the other:

- *SI_Movie_1*: chaotic bands, $\kappa_\beta^c = -\kappa_\alpha^c = 0.1$, $\kappa_\beta^c = -\kappa_\alpha^c = 0.9$.
- *SI_Movie_2*: steady band, $\kappa_\beta^c = -\kappa_\alpha^c = 0.9$, $\kappa_\beta^c = -\kappa_\alpha^c = 0.1$.
- *SI_Movie_3*: intermittent dynamical behavior, $\kappa_\alpha^c = -0.1$, $\kappa_\beta^c = 0.3$.
- *SI_Movie_4*: $\kappa_\alpha^c = -1$, $\kappa_\beta^c = 0.1$.
- *SI_Movie_5*: $\kappa_\alpha^c = -0.3$, $\kappa_\beta^c = 0.9$.

To produce *SI Movie 6*, we instead use both pairwise forces and quorum sensing, as explained in Sec. V. We use a system size of 75×75 , $dt = 0.05$, $\bar{\rho} = 0.50$, $\delta\rho = 0.01$, $v_{\alpha,\beta}^0 = 0.5$, $\tau_{\alpha,\beta}^{-1} = 0.018$, $r_0 = 0.89$, $E_0 = 30$, and $r_{QS} = 5$. The simulations are initialized in a homogeneous configuration with densities $\rho_\alpha^0, \rho_\beta^0$. To explore the static and dynamical phenomenology of the system, we vary both the initial densities and the strength of cross-interactions κ_μ^c . *SI Movie 6* comprises four panels, whose parameters are both reported in the Movie and summarized in the table below.

Movie	Phenomenology	ρ_α^0	ρ_β^0	κ_α^c	κ_β^c
SI_Movie_6 panel a	Collective MIPS	0.40	0.40	0	0
SI_Movie_6 panel b	Demixing	0.40	0.40	4	4
SI_Movie_6 panel c	Triple-coexistence	0.40	0.50	4	4
SI_Movie_6 panel d	Dynamical	0.60	0.60	4	-4

~

VII. SUPPLEMENTARY FIGURES

Figure S1. Pictorial representation of the lever rule in the triple-triple-phase region. The initial composition of our system ρ^0 can be seen as the center of mass of three particles with mass $1 - s - t$, s , t located at ρ^1 , ρ^2 , ρ^3 respectively. In this representation, the relative weight of each phase is then given by the value of its corresponding mass.

Figure S2. Chaotic bands in the absence of self-interaction $\kappa_{\alpha,\beta}^s = 0$. Parameters: $\kappa_{\alpha}^c = -\kappa_{\beta}^c = -1$, $\rho_{\alpha}^0 = \rho_{\beta}^0 = 50$, $\bar{\rho} = 50$, $\delta\rho = 20$, $v_{\alpha,\beta}^0 = 5$, $\tau_{\alpha,\beta} = 1$, system size 30×30 , $dt = 0.005$.

~

-
- [1] M. E. Cates and J. Tailleur, *Europhysics Letters* **101**, 20010 (2013).
 - [2] D. Martin, J. O'Byrne, M. E. Cates, É. Fodor, C. Nardini, J. Tailleur, and F. Van Wijland, *Physical Review E* **103**, 032607 (2021).
 - [3] H. Ehrentraut and W. Muschik, *ARI-An International Journal for Physical and Engineering Sciences* **51**, 149 (1998).
 - [4] J. A. Schouten, *Tensor analysis for physicists* (Courier Corporation, 1989).
 - [5] A. Spencer, *International Journal of Engineering Science* **8**, 475 (1970).
 - [6] A. P. Solon, M. E. Cates, and J. Tailleur, *The European Physical Journal Special Topics* **224**, 1231 (2015).
 - [7] D. S. Dean, *Journal of Physics A: Mathematical and General* **29**, L613 (1996).
 - [8] A. Kriegl and P. W. Michor, *The convenient setting of global analysis*, Vol. 53 (American Mathematical Soc., 1997).
 - [9] R. Brunetti, K. Fredenhagen, and P. L. Ribeiro, *Communications in Mathematical Physics* **368**, 519 (2019).
 - [10] H. Cartan, *Differential calculus*. hermann (1971).
 - [11] J. Dieudonné, *Foundations of modern analysis* (Read Books Ltd, 2011).
 - [12] S. Arora, H. Browne, and D. Daners, *Journal of the Australian Mathematical Society* **111**, 202 (2021).
 - [13] J. O'Byrne and J. Tailleur, *Physical Review Letters* **125**, 208003 (2020).
 - [14] C. Nardini, É. Fodor, E. Tjhung, F. Van Wijland, J. Tailleur, and M. E. Cates, *Physical Review X* **7**, 021007 (2017).
 - [15] U. Seifert, *Physical Review Letters* **95**, 040602 (2005).
 - [16] F. Otto, *Communications in Partial Differential Equations* **26**, 101 (2001).
 - [17] J. O'Byrne, *Physical Review E* **107**, 054105 (2023).
 - [18] P. Sollich, *Journal of Physics: Condensed Matter* **14**, R79 (2001).
 - [19] A. P. Solon, Y. Fily, A. Baskaran, M. E. Cates, Y. Kafri, M. Kardar, and J. Tailleur, *Nature Physics* **11**, 673 (2015).

REVIEWER COMMENTS

Reviewer #1 (Remarks to the Author):

I want to thank the authors for considering my comments, which were satisfactorily addressed in their replies. I support the publication of the manuscript on Nature Communications in its present form.

Reviewer #2 (Remarks to the Author):

I thank the authors for their careful revision of the manuscript. They have added new material to make the content more general, they have added the definition of σ in the manuscript to make it more self-contained, they have added more explanations to old Eqs. (10) and (14), and the explanation of Fig. 1 has been strongly improved.

I have two points, which I ask the authors to reconsider again, then I would be happy to recommend the manuscript for publication.

1. The authors now argue a lot with bacterial systems for what they present (quorum sensing and chemotaxis). Therefore, I suggest two make the title more specific, writing, for example, "Non-reciprocity across scales in bacterial mixtures".

2. About steady state: The example, which the authors present, is exactly what I meant: the relaxation of the particle distribution into some steady-state distribution. As I understand, this is not included in the cases to which the entropy production rate in new Eq.(7) applies. Correct? So, when their coarse-grained equations describe time-dependent patterns, can σ make any predictions for these patterns? Or do the authors need to involve some time-scale separation, where the relaxation of the distribution is short compared to the characteristic time scales of the patterns. It would be good to add some further clarifications here.

Reviewer #3 (Remarks to the Author):

The authors have provided some satisfactory answers to all the comments raised by the Referees. Therefore, I recommend publication of the manuscript in its present form.

Report of reviewer #2

I thank the authors for their careful revision of the manuscript. They have added new material to make the content more general, they have added the definition of σ in the manuscript to make it more self-contained, they have added more explanations to old Eqs. (10) and (14), and the explanation of Fig. 1 has been strongly improved.

Response: We thank the referee for these positive comments.

I have two points, which I ask the authors to reconsider again, then I would be happy to recommend the manuscript for publication.

1. The authors now argue a lot with bacterial systems for what they present (quorum sensing and chemotaxis). Therefore, I suggest to make the title more specific, writing, for example, "Non-reciprocity across scales in bacterial mixtures".

Response: We respectfully disagree with the suggestion of the referee. As mentioned in several places in our article, our results hold beyond the case of bacterial mixtures, which is used mostly for illustration purpose:

- Quorum sensing and chemotaxis are generically experienced by other types of cells.
- Mediated interactions are also relevant for self-propelled colloids interacting via diffusiophoresis and our results would extend directly to that case.
- Our results directly extend to ABPs and AOUPs.

Overall, we fear that speaking about bacteria in the title may discourage readers interested in active matter but not in biophysics from reading our article. Clearly, our results on the fate of non-reciprocity across scales are more general than the case of bacterial mixtures.

2. About steady state: The example, which the authors present, is exactly what I meant: the relaxation of the particle distribution into some steady-state distribution. As I understand, this is not included in the cases to which the entropy production rate in new Eq.(7) applies. Correct?

Response: This is correct and rather well explained in the 2005 Seifert PRL (ref 63). The relaxation from an initial measure $\mathbf{P}_0[\rho]$ to a steady-state measure $\mathbf{P}_g[\rho]$ leads to a change of Gibbs-Shannon entropy, which is not captured by the steady-state entropy production rate σ (by definition). The latter instead captures solely the non-trivial, time-extensive entropy production that occurs in nonequilibrium steady states.

We thank the referee for this question and now state this explicitly, right after the definition of σ , so that readers who are not familiar with stochastic thermodynamics can more easily appreciate what σ captures.

So, when their coarse-grained equations describe time-dependent patterns, can σ make any predictions for these patterns? Or do the authors need to involve some time-scale separation, where the relaxation of the distribution is short compared to the characteristic time scales of the patterns. It would be good to add some further clarifications here.

Response: First, we stress that σ measures the steady-state irreversibility of the dynamics irrespective of the nature of the steady-state distribution $\mathbf{P}_g[\{\rho_v\}]$.

When $\mathbf{P}_g[\{\rho_v\}]$ is dominated by static patterns, the steady-state dynamics of the fields ρ_v corresponds to fluctuations around these static patterns. A positive value of σ then implies that the dynamics of these fluctuations are irreversible.

When $\mathbf{P}_g[\{\rho_v\}]$ is dominated by traveling patterns, the steady-state dynamics of ρ include both

the average dynamics of the patterns and the fluctuations around them. The positive nature of \mathbf{a} then stems both from the irreversible dynamics of the patterns and from the irreversibility of the fluctuations around them. (It would actually be interesting to see how \mathbf{a} changes across a transition from static to dynamical patterns, but that's another research project!).

In short, whether the steady-state is dominated by static structures or dynamical ones, \mathbf{a} measures the full irreversibility of the coarse-grained dynamics. We now mention this the first time we discuss traveling patterns and we have also added a footnote later on with references to the relevant literature.

All in all, we hope that the referee will accept our desire to have a broad title and that our explanations regarding \mathbf{a} will satisfy their curiosity.